# BAT-CLIP: BIMODAL TEST-TIME ADAPTATION FOR CLIP

## ABSTRACT

Although open-vocabulary classification models like Contrastive Language Image Pretraining (CLIP) have demonstrated strong zero-shot learning capabilities, their robustness to common image corruptions remains poorly understood. Through extensive experiments, we show that zero-shot CLIP lacks robustness to common image corruptions at increasing severity levels during test time, necessitating the adaptation of CLIP to unlabeled corrupted images using test-time adaptation (TTA). However, we found that existing TTA methods have severe limitations in adapting CLIP due to their *unimodal* nature. To address these limitations, we propose **BAT-CLIP**, a *bimodal* TTA method specially designed to improve CLIP's robustness to common image corruptions. The key insight of our approach is not only to adapt the visual encoders for better image feature extraction but also to strengthen the alignment between image and text features by promoting a stronger association between the image class prototype, computed using pseudo-labels, and the corresponding text feature. We evaluate our approach on benchmark image corruption datasets and achieve state-of-the-art results in TTA for CLIP, specifically for domains involving image corruptions. Particularly, with a ViT-B/16 vision backbone, we obtain mean accuracy improvements of 9.7%, 5.94%, and 5.12% for CIFAR-10C, CIFAR-100C, and ImageNet-C, respectively.

## 1 INTRODUCTION

The emergence of large pre-trained vision-language models (VLMs), such as CLIP (Radford et al., 2021), has led to their widespread adoption in various visual recognition tasks, including segmentation (Li et al., 2022; Luo et al., 2023), detection (Bangalath et al., 2022; Lin & Gong, 2023), classification (Zhou et al., 2022b;a), and image generation (Vinker et al., 2022; Ramesh et al., 2022; Rombach et al., 2022). Thanks to supervision from massive corpora of paired language and image data, VLMs like CLIP demonstrate strong zero-shot capabilities for these downstream tasks.

Despite CLIP's successes in such important applications, its robustness when faced with corrupted images remains largely underexplored. Our motivation stems from the fact that the vision perception system of humans exhibits a level of robustness that real-world vision systems are yet to achieve. For example, models deployed for safety-critical applications like autonomous driving (Arnold et al., 2019), could face rapid distributional shifts of blurriness, pixel changes, snowy nights, or other weather conditions (Sakaridis et al., 2021). For instance, our findings on the zero-shot performance of CLIP with a ResNet-101 (He et al., 2016) vision backbone reveals that the accuracy on the test set of CIFAR100 (Krizhevsky et al., 2009) with *Gaussian* noise of severity level 5, plummets to 10.79% from 49% on the clean set. Similar trends are observed with ViT-B/16, -B/32, and -L/14 (Dosovitskiy, 2020) as backbones. Behaviors such as these could lead to severe performance degradation of models when faced with image corruption in real-world scenarios.

This challenge is not unique to CLIP; it reflects a broader issue in computer vision tasks (Deng et al., 2009; Everingham et al., 2010; Faster, 2015; Chen et al., 2017), which often rely on the assumption that training and test data share the same distribution. When test distributions differ, model adaptation becomes essential to accommodate this shift and maintain accurate predictions. In response to this issue, test-time adaptation (TTA) has garnered significant interest (Wang et al., 2021; Döbler et al., 2023). TTA aims to adapt a pre-trained model to unlabeled batches of test data from varying domains in an online fashion—updating the model for each test batch without

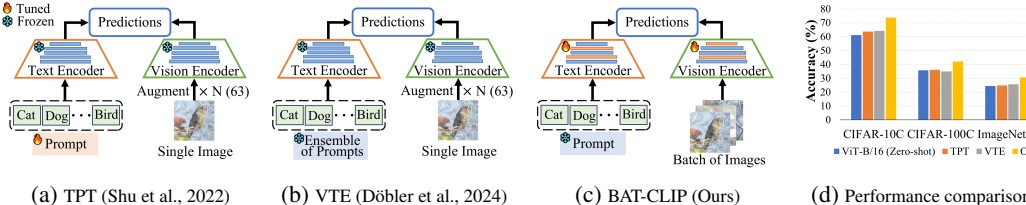

(a) TPT (Shu et al., 2022)  (b) VTE (Döbler et al., 2024)  (c) BAT-CLIP (Ours)  (d) Performance comparison

Figure 1: Comparison of **BAT-CLIP** with other TTA approaches using CLIP: TPT (Shu et al., 2022), and VTE (Döbler et al., 2024). a) TPT optimizes text prompts only for a single test image, making it *unimodal*. b) VTE considers an ensemble of prompts without model updates. Both methods consider the generation of multiple augmentations of the test image. c) BAT-CLIP is a *bimodal* approach, that adapts the LayerNorm parameters of the vision and text encoders, maximizing alignment between class prototypes and text features while increasing the inter-class separability of prototypes.

access to the source dataset. Several TTA methods proposed in the literature have been effective in mitigating domain shifts (Wang et al., 2021; Schneider et al., 2020; Rusak et al., 2021; Niu et al., 2022; Sun et al., 2020; Chen et al., 2022). In this paper, we use the terms distributions and domains interchangeably.

While there have been few new works on using CLIP for TTA (Shu et al., 2022; Döbler et al., 2024), they come with certain limitations. For example, test-time prompt tuning (TPT) (Shu et al., 2022) tunes the text prompts on the text encoder alone and generates multiple random augmented views, for each test image. The text prompts, initialized to pre-trained values, are optimized by minimizing the marginal entropy of the confident model predictions. The prompts are reset after adaptation to each image. However, such a method is expensive and slow due to performing multiple forward passes through the vision encoder of CLIP, for each image. Also, it relies on hand-crafted prompts for initialization, making it impractical at test-time. Another approach, Vision-Text-Space Ensemble (VTE) (Döbler et al., 2024), uses an ensemble of different prompts as input to CLIP's text encoder while keeping the encoders frozen. However, since the vision encoder remains frozen, it struggles to adapt images with severe noise effectively.

While TPT effectively improves CLIP's test generalization by dynamically tuning the text prompts, it remains primarily a *unimodal* approach. This limits the capacity of a multi-modal model like CLIP to fully leverage its multi-modal nature for adaptation. Specifically, it prevents the encoders from jointly adjusting their features, resulting in suboptimal alignment between the visual and text modalities. During TTA, there is no transfer of knowledge, via gradients, between the visual and text encoders, as the adaptation process focuses solely on one modality. For instance, when a test image includes common corruptions, the text prompts adapt, but the vision encoder features remain fixed. As a result, the learned prompts lack awareness of the test image distribution, leading to a less effective adaptation.

To address these core limitations, we propose, **BAT-CLIP**, a *bimodal* adaptation approach of CLIP for TTA, where both the visual and text encoders are jointly adapted by exploiting CLIP's shared feature space of images and text. The overall objective is to achieve a strong alignment between image features and text features to enable more effective multi-modal learning and adaptation. The adaptation procedure is two-fold: 1) Similar to TENT (Wang et al., 2021), we adapt the norm layers *i.e.,* LayerNorm layers of both encoders. However, such a model update does not consider the alignment of the encoder features. So, to improve the alignment between class-specific visual and text features, we introduce a projection matching loss that maximizes the projection of the visual class prototypes with their corresponding text features. 2) To learn more discriminative visual features, we increase the cosine distance between the class prototypes, promoting a more distinct separation in the image feature space. Furthermore, our method is general-purpose, as it does not rely on hand-crafted prompt templates or their ensembles, unlike VTE (Döbler et al., 2024). We leverage batches of test samples for TTA, rather than focusing on single-image adaptation - making our method fast and efficient. In Fig. 7, we show several examples of classification results, as a comparison.

To the best of our knowledge, the proposed method is the first to perform *bimodal* test-time adaptation (TTA) of CLIP for classification tasks. We draw a comparison of our method against TPT and VTE in Fig. 1. Our main contributions are as follows:

- We begin by conducting a comprehensive analysis of CLIP's zero-shot performance, for various visual backbones, on common image corruptions, at test-time, with varying levels of severity. We observed that while CLIP demonstrates strong zero-shot performance on clean images, its performance declines significantly when handling corrupted images.

- To address the *unimodal* limitations highlighted earlier, our proposed *bimodal* adaptation of CLIP encoders at test-time aims to enhance alignment by maximizing the projection of class-wise prototypes onto their corresponding text features. Simultaneously, we increase the cosine distance between class prototypes to encourage the learning of more discriminative features, making the test adaptation process more flexible and robust.

- We conduct extensive experiments and benchmark our results against others using CLIP for TTA. We also adopt prior TTA approaches to use CLIP and compare them against ours. Our method results in the state-of-the-art for CLIP test adaptation to common corruptions.

## 2 RELATED WORKS

**Online Test-Time Adaptation (TTA).** The objective of TTA is to adapt a pre-trained model, with no access to the source data, to incoming batches of unlabelled test data of a specific domain (Wang et al., 2021; Sun et al., 2020; Schneider et al., 2020; Rusak et al., 2021; Niu et al., 2022; Chen et al., 2022; Zhang et al., 2022; Choi et al., 2022). In this approach, the model is reset to its source-domain pre-trained state after adapting to each target domain. As the updates are performed online, TENT (Wang et al., 2021) updates the affine parameters of the normalization layers and minimizes the entropy (Shannon, 1948) of the model predictions. BN Stats Adapt (BN-1) (Schneider et al., 2020) changes only the statistics of the normalization layers with those of the test batch to reduce the covariate shift due to corruptions. RPL (Rusak et al., 2021) argues that self-learning during adaptation via entropy minimization and pseudo-labels is beneficial. They propose the usage of a generalized cross-entropy loss for adaptation. SAR (Niu et al., 2022) filters noisy test samples that cause a performance drop identified from the gradient space. However, none of these works utilize CLIP for TTA.

**TTA using CLIP.** Lately, CLIP has been finding extensive applications for TTA. TPT (Shu et al., 2022) was the first work to propose prompt tuning using CLIP at test-time. However, this method is computationally intensive for generating multiple views, per image. Additionally, a key limitation of this approach is its reliance on using CLIP's default hand-crafted template for prompt initialization, with resets after adaptation to each image. A similar line of work was done in VTE (Döbler et al., 2024) where ensembles are created in the text and vision space without any CLIP parameter update. Sreenivas & Biswas (2024) focus on single-image TTA for out-of-distribution detection and test generalization (Li et al., 2023; Lee et al., 2023). While TPT is *unimodal*, our approach is *bimodal*, involving the joint optimization of both encoders for test adaptation, leading to a strong multi-modal alignment between features. We employ a single generic prompt template, eliminating the need for prompt engineering and making our method more suitable for real-time deployment.

## 3 ZERO-SHOT PERFORMANCE ANALYSIS OF CLIP TO COMMON IMAGE CORRUPTIONS

While CLIP generalizes well to new concepts across vision-language modalities (Chen et al., 2021; Han et al., 2021), its performance under image corruptions is less explored. This section evaluates CLIP's zero-shot capabilities in real-world scenarios with domain shifts caused by common corruptions, focusing on two primary areas: zero-shot performance and the need for adaptation to address domain shifts effectively.

**Vision Backbones.** We evaluate the robustness with a ResNet-101 (RN101) vision backbone (He et al., 2016) and three Vision Transformer backbones (ViT-B/16, ViT-B/32, ViT-L/14) (Dosovitskiy, 2020). The models are tested on their zero-shot classification performance.

**Datasets.** For all the experiments in this paper, we utilize the CIFAR-10C, CIFAR-100C, and ImageNet-C datasets (Hendrycks & Dietterich, 2019), each containing 15 distinct corruption types as tasks (*e.g.,* Gaussian noise, Shot noise, Impulse Noise, Defocus Blur, etc.). Each corruption is applied at 5 different severity levels to the test sets of CIFAR10, CIFAR100 (Krizhevsky et al., 2009),

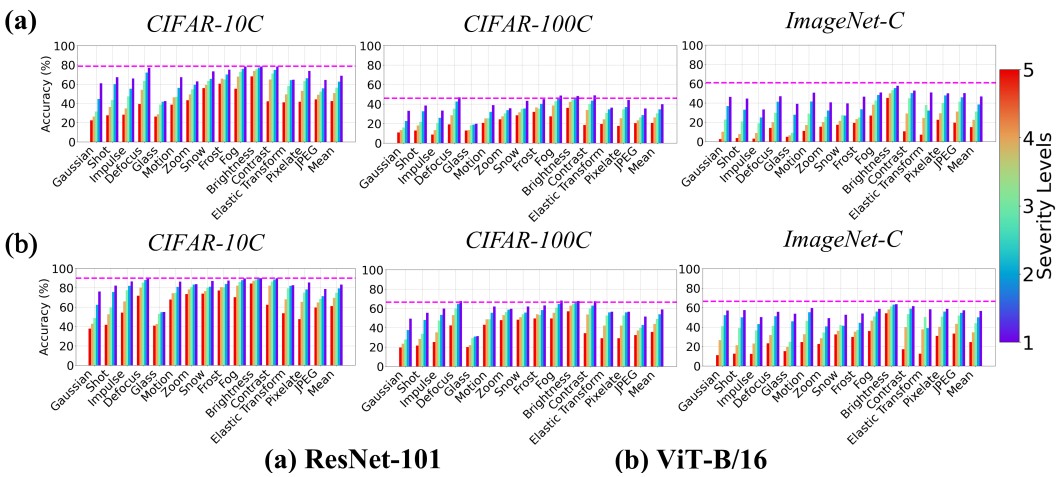

Figure 2: Task-wise mean accuracy (%) of zero-shot CLIP across different corruption severity levels. [Top]: ResNet-101 backbone. [Bottom]: ViT-B/16 backbone. The **dashed lines** indicate the performance of zero-shot CLIP (w/ respective visual backbones) on the corresponding source test sets.

and ImageNet (Deng et al., 2009), acting as source test sets, and allowing us to systematically evaluate the model's performance under increasing degrees of image degradation. We provide additional dataset details in Appendix A.1.

**TTA Problem Setup.** Each corruption type of a certain severity level, posed as a task $\mathcal{T}_i$ with $\mathcal{B}$ test batches, is sequentially presented to CLIP's vision encoder for model predictions, with each test batch being revealed one at a time. Let a batch of images from task $\mathcal{T}_i$, at time step $t$, be denoted as $x_i^t$. For the prompt template, unless explicitly mentioned, we always use the generic "a photo of a <CLS>." to generate $C$ text representations ($\mathcal{Z} = \{z_c\}_{c=1}^C$), where $C$ is the total number of classes. Let $f_{vis}$ and $f_{txt}$ denote CLIP's vision and text encoder, respectively. The visual feature of the $k^{th}$ image in batch $x_i^t$ is $v_{k,i}^t = f_{vis}(x_{k,i}^t)$. The likelihood of it belonging to class $c$ is,

$$p(y = c | x_{k,i}^t) = \frac{\exp(\text{sim}(v_{k,i}^t, z_c)/\tau)}{\sum_j \exp(\text{sim}(v_{k,i}^t, z_j)/\tau)}; \quad \text{sim}(v, z) = \frac{v^T \cdot z}{||v||_2 \cdot ||z||_2} \qquad (1)$$

where $\text{sim}(\cdot)$ is the cosine similarity and $\tau$ is the softmax temperature from CLIP's pre-training stage. The text features $\mathcal{Z}$, in zero-shot evaluation, are always pre-computed. We also draw a comparison with CLIP performance on respective source test sets and follow [1] for this implementation. Throughout this paper, the same TTA problem setup is employed.

### 3.1 SENSITIVITY OF CLIP TO IMAGE CORRUPTION SEVERITY

We analyze CLIP's zero-shot performance by progressively increasing corruption severity for RN101 and ViT-B/16 visual backbones. We report mean accuracy and overall performance in Fig. 2. Despite CLIP's robust multimodal feature space, accuracy drops significantly with an increase in corruption severity, regardless of the backbone. Specifically, for CIFAR-10C with a ViT-B/16 backbone, we observe accuracy as low as 37.92%, at a severity level of 5, for *Gaussian* noise. Similarly, for CIFAR-100C and ImageNet-C, the mean accuracy rates are as low as 35.79% and 24.51% at a severity level of 5, respectively. Results on additional vision backbones are in Appendix A.3.1

**Analysis.** CLIP's zero-shot classification accuracy varies across models and datasets. Comparing the results to source test sets, for CIFAR10, RN101 achieves 78.8%, improving to 90.1% with ViT-B/16. On CIFAR100, accuracies are 46.1% and 66.6%, respectively. For ImageNet, RN101 scores 61.2%, while ViT-B/16 reaches 67.7%. More importantly, the key takeaway from Fig. 2 is — even a slight increase in severity to level 1, for a majority of the corruption tasks, leads to a noticeable drop

---

[1]https://github.com/LAION-AI/CLIP_benchmark

Table 1: Mean classification accuracy (%) across all corruption tasks with different prompt templates. For each, we report the drop in accuracy (−) (in %) compared to the performance on the corresponding source test set.

| Prompt Template | Backbone | CIFAR-10C | CIFAR-100C | ImageNet-C |
|---|---|---|---|---|
| "a photo of a <CLS>." | RN101 | 42.30 (−36.50) | 20.62 (−25.48) | 14.97 (−46.23) |
| | ViT-B/16 | 61.16 (−28.84) | 35.78 (−30.82) | 24.51 (−42.16) |
| | ViT-B/32 | 59.00 (−29.20) | 31.79 (−30.51) | 23.20 (−38.80) |
| | ViT-L/14 | 75.84 (−19.36) | 47.82 (−27.78) | 39.55 (−33.95) |
| "a bad photo of a <CLS>." | RN101 | 42.73 (−36.47) | 20.18 (−27.02) | 12.87 (−49.43) |
| | ViT-B/16 | 62.97 (−28.03) | 36.37 (−30.33) | 25.84 (−42.26) |
| | ViT-B/32 | 60.25 (−29.75) | 31.30 (−31.00) | 23.58 (−39.22) |
| | ViT-L/14 | 76.08 (−19.22) | 48.09 (−25.41) | 39.37 (−34.73) |
| "a blurry photo of a <CLS>." | RN101 | 45.32 (−34.98) | 20.28 (−26.12) | 15.80 (−45.70) |
| | ViT-B/16 | 62.49 (−28.31) | 35.17 (−30.83) | 25.27 (−42.03) |
| | ViT-B/32 | 57.85 (−30.65) | 31.47 (−30.85) | 23.40 (−38.80) |
| | ViT-L/14 | 73.77 (−20.73) | 48.07 (−25.73) | 39.22 (−34.18) |
| "a noisy photo of a <CLS>." | RN101 | 44.56 (−34.34) | 20.71 (−25.89) | 15.48 (−45.72) |
| | ViT-B/16 | 63.03 (−27.87) | 35.14 (−30.76) | 25.05 (−41.35) |
| | ViT-B/32 | 59.76 (−29.24) | 31.53 (−30.37) | 23.11 (−38.19) |
| | ViT-L/14 | 75.36 (−19.54) | 47.87 (−27.63) | 38.78 (−34.42) |

in accuracy. One plausible explanation for CLIP's subpar performance is that the parameters of $f_{vis}$ were not optimized for such corruptions during pre-training. In zero-shot classification, the visual features from a given domain may lack the robustness and richness necessary to align well with their corresponding text features. This results in lower likelihoods and thus, higher misclassification rates.

## 3.2 SENSITIVITY OF CLIP TO PROMPT TEMPLATES

In this analysis, we evaluate the impact of prompt engineering by providing "relevant" prompt templates to $f_{txt}$ for TTA. Each prompt adds context to help CLIP extract more relevant text features. We report the mean classification accuracy (in %) across RN101, ViT-B/16, ViT-B/32, and ViT-L/14 backbones at an image corruption severity level of 5 for all datasets, with results summarized in Table 1. The absolute accuracy drop compared to zero-shot performance on the source test set is also reported.

**Analysis.** It is interesting to observe that, irrespective of the backbones used, we do not see any drastic changes in the mean accuracy, for different "relevant" prompt templates. However, the major concern arises in the performance gap of each model and the zero-shot CLIP performance on the corresponding source test set, for the same prompt template. This discrepancy highlights the limited robustness of CLIP's text encoder $f_{txt}$ to prompt selection in the context of image corruption. A key reason for this is that, despite the use of "relevant" prompts, the text and visual features remain largely independent and unaware of one another.

As expected, RN101 performs worse than the ViT-based backbones, primarily due to its lack of global attention-based modeling inherent to transformers (Vaswani, 2017). Therefore, for the remainder of the experiments in this paper, we focus on ViT-based backbones, specifically ViT-B/16 and ViT-B/32. Due to GPU memory limitations, we do not use ViT-L/14 for model adaptation.

**Unsuitability of prompt template selection at test-time.** Additionally, at test-time, it is impractical to perform prompt engineering or optimize prompt vectors since 1) Choosing different prompt templates for generating text features is extremely tedious and time-consuming. 2) In real-time deployment involving prompt-tuning, such prompts cannot quickly estimate the distribution of incoming test batches. TPT (Shu et al., 2022) optimizes pre-trained text prompts for each test image, turning out to be suboptimal since the prompts are optimized ignorant of the distribution of the test image.

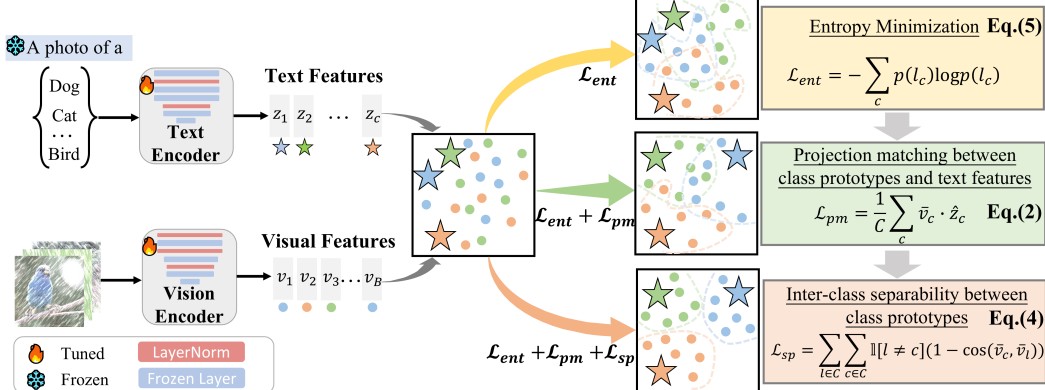

Figure 3: **BAT-CLIP** not only adapts the visual encoder for highly discriminative image features but also promotes a strong alignment between image and text features by adapting the text encoder too, leading to improved performance following test-time adaptation. We adapt only the LayerNorm parameters of CLIP encoders.

## 4  BAT-CLIP

The comprehensive analysis in Section 3 reveals that the zero-shot vision encoder $f_{vis}$ of CLIP is very sensitive to image corruption with increasing severity. Similarly, the performance of the text encoder $f_{txt}$ is invariant to the different text prompt templates and is also impractical to tune at test-time. Therefore, for efficient adaptation, both the image and text features of CLIP need to be *adapted* to the incoming domain of test batches. We illustrate our proposed framework, **BAT-CLIP**, in Fig. 3.

**From unimodal adaptation to bimodal adaptation.** The goal of TTA is to enhance a model's performance in the current domain to ensure accurate predictions. To handle the complexities of CLIP adaptation to a specific domain of image corruption, we dissect our analysis of each encoder's adaptation. Consider the *unimodal* adaptation of the vision encoder $f_{vis}$ via entropy minimization (Wang et al., 2021). While the image features are adjusted for a specific test batch and domain, the text features remain fixed, still being optimized for the data from CLIP pre-training, as seen in Fig. 3 (top row). Likewise, if only the text encoder $f_{txt}$ is updated in this manner, the generated text features may not align properly with the distribution of the incoming data, potentially causing a misalignment between the image and text features.

To benefit from the feature space and learn richer representations across both modalities, we propose adapting both $f_{vis}$ and $f_{txt}$ to a domain, enabling an input-aware knowledge transfer between the encoders and enhancing domain-specific adaptation. While the encoders can be updated based on entropy minimization, they come with inherent limitations. Though entropy models the prediction uncertainty, it does not guarantee increasing the likelihood of alignment between the image and text features. Additionally, due to the image corruption, for images within a test batch, there could be a possible overlap of visual features belonging to different classes (Kurakin et al., 2016). The robustness of CLIP would be challenged in such a case. To address these limitations, we propose two loss components that facilitate more efficient and effective *bimodal* adaptation of CLIP to new domains at test-time. The losses focus on maximizing alignment between the visual and text features while increasing separation between the visual features to learn good decision boundaries.

**Projection matching between the visual and text features.** We propose learning visual and text features that are domain-specific and mutually aware by *jointly updating the encoders*, as in Fig.3 (middle row). From Eq. 1, the visual feature $v_{k,i}^t$ of the $k^{th}$ image in batch $x_i^t$ needs to have a high similarity with the text feature $z_c$ of class $c$ for good alignment. To quantify this similarity, a possible direction is to project the visual feature onto the text feature *i.e.,* compute the scalar projection — $v_{k,i}^t \cdot \hat{z}_c$, where $\hat{z}_c$ is the normalized text feature of $\hat{z}_c$ *i.e.,* $\hat{z}_c = \frac{z_c}{||z_c||_2}$. Geometrically speaking, maximizing this projection leads to more similarity between the corresponding features and hence, a better alignment.

A possible approach would be to compute the projection for each class-specific visual feature in a test batch. However, within the batch, due to the image corruption, the predicted labels could be wrong/noisy. Instead, we propose modeling the projection of the class prototype with its corresponding text feature. A prototype is useful, in such a scenario, because it encompasses the entire class distribution without relying on individual visual features (Snell et al., 2017). In particular, we compute a class prototype as,

$$\bar{v}_c = \frac{1}{\sum_{k=1}^{\mathcal{B}} \mathbb{1}[\hat{y} = c]} \sum_{k=1}^{\mathcal{B}} \mathbb{1}[\hat{y} = c] v_{k,i}^t; \quad \hat{y} = \underset{c}{\operatorname{argmax}} \, p(y|x_i^t) \qquad (2)$$

*i.e.*, we compute the mean feature of all the support visual features constituting a class $c$. $\hat{y}$ refers to the predicted labels computed via Eq. 1 and $\bar{v}_c$ is the class prototype of class $c$. Based on the class prototype, the projection matching loss is,

$$\mathcal{L}_{pm} = \frac{1}{C} \sum_c \bar{v}_c \cdot \hat{z}_c \qquad (3)$$

Eq. 3 encourages a class prototype to have a larger projection on its corresponding text feature. In this way, during the adaptation of CLIP to a certain domain, both encoders learn to generate richer visual and text features with maximum alignment by maximizing Eq. 3. Fig. 3 (middle row) shows such an illustration.

**Inter-class separability between class prototypes.** The projection matching loss introduced in Eq. 3 encourages the $f_{vis}$ and $f_{txt}$ encoders to produce domain-specific, well-aligned, and input-aware features via jointly updating the encoders. However, when TTA occurs at a batch level with image corruption, visual features within the batch could overlap, leading CLIP to poorly differentiate between classes. This ultimately hinders effective class separation. Hence, with a desire to obtain separation between visual features from different classes, as illustrated in Fig. 3 (last row), we propose maximizing the distance between the class prototypes. Since the visual and text features align across modalities via Eq. 3, class separation, in addition, is needed for good generalization, robustness, and adaptation. So, we increase the cosine distance between the class prototypes and enhance the discriminative nature as,

$$\mathcal{L}_{sp} = \sum_{l \in C} \sum_{c \in C} \mathbb{1}[l \neq c](1 - \cos(\bar{v}_c, \bar{v}_l)) \qquad (4)$$

**Optimization.** TENT (Wang et al., 2021) optimizes the output logits by minimizing the entropy Shannon (1948). When applied to CLIP, the entropy, with output logits $l$, is defined as,

$$\mathcal{L}_{ent} = -\sum_c p(l_c) \log p(l_c) \qquad (5)$$

where $p(l_c)$ is the likelihood for class $c$ that is computed via Eq. 1. The overall optimization objective for our approach is as,

$$\underset{\phi_v, \phi_t}{\operatorname{argmin}} (\mathcal{L}_{ent} - \mathcal{L}_{pm} - \mathcal{L}_{sp}) \qquad (6)$$

where $\phi_v$ and $\phi_t$ refer to the parameters of the vision and text encoder, respectively. We update only the *LayerNorm parameters* of $f_{vis}$ and $f_{txt}$, as outlined by the critical analysis provided by Sreenivas & Biswas (2024). This constitutes updating $\sim 0.044\%$ of all CLIP model parameters. For every new task, we reset the model parameters of CLIP following TENT (Wang et al., 2021) since our goal is to adapt to a single domain in an **online** manner. We use a generic hand-crafted prompt template "a photo of a <CLS>." at all times, thus avoiding the need for a prompt search. In summary, our *bimodal* test-time adaptation approach jointly updates the LayrNorm affine parameters of the CLIP encoders that are optimized synergistically through loss components aware of the input domain, leading to a more robust multimodal learning process.

## 5 Experiments and Results

### 5.1 Online Test-Time Adaptation

**Baselines.** We compare our approach with zero-shot CLIP and other proposed TTA methods using CLIP - TPT (Shu et al., 2022) and VTE (Döbler et al., 2024). TPT was the first work to propose

Table 2: Mean accuracy (%) on CIFAR-10C, CIFAR-100C, and ImageNet-C - TTA mean accuracy of the 15 corruptions (tasks) at a severity level of 5, using ViT-B/16 and ViT-B/32. **+** and **-** denote the absolute gain/loss *w.r.t* the next best performance. We contrast our results against zero-shot ViT-B/16 and /32, TPT (Shu et al., 2022), and VTE (Döbler et al., 2024).

| | Method | Gaussian | Shot | Impulse | Defocus | Glass | Motion | Zoom | Snow | Frost | Fog | Brightness | Contrast | Elastic | Pixelate | JPEG | Mean |
|---|---|---|---|---|---|---|---|---|---|---|---|---|---|---|---|---|---|
| **CIFAR-10C** | ViT-B/16 | 37.92 | 41.7 | 54.42 | 71.75 | 40.89 | 67.93 | 73.62 | 73.89 | 77.35 | 70.22 | 84.45 | 62.36 | 53.81 | 47.65 | 59.43 | 61.16 |
| | TPT | 37.74 | 42.24 | 60.57 | 72.88 | 44.80 | 69.69 | 75.37 | 75.96 | 78.84 | 72.12 | 85.68 | 62.04 | 58.90 | 55.14 | 62.64 | 63.64 |
| | VTE | 42.42 | 46.26 | 64.23 | 71.10 | 45.58 | 68.50 | 73.66 | 76.75 | 78.27 | 71.02 | 85.28 | 57.24 | 59.54 | 60.59 | 61.85 | 64.15 |
| | Ours | **61.13** | **64.09** | **65.76** | **80.51** | **54.96** | **80.65** | **81.94** | **83.04** | **84.19** | **80.84** | **88.95** | **82.15** | **69.16** | **62.68** | **66.64** | **73.85** |
| | Gain/Loss(%) | +18.71 | +17.83 | +1.53 | +7.63 | +9.38 | +10.96 | +6.57 | +6.29 | +5.35 | +8.72 | +3.27 | +19.79 | +9.62 | +2.09 | +4.00 | +9.70 |
| | ViT-B/32 | 35.47 | 39.94 | 43.23 | 69.95 | 41.43 | 64.50 | 70.13 | 70.85 | 72.33 | 66.66 | 81.37 | 64.57 | 59.69 | 48.28 | 56.62 | 59.00 |
| | TPT | 43.11 | 46.53 | 48.29 | 71.31 | 47.80 | 66.89 | 71.96 | 74.00 | 76.00 | 68.81 | 84.12 | 66.35 | 63.86 | 51.86 | 58.01 | 62.59 |
| | VTE | 47.59 | 50.18 | **53.15** | 71.39 | 53.86 | 67.92 | 72.90 | 76.37 | 76.30 | 70.78 | 83.27 | 61.07 | **69.00** | **58.57** | 61.14 | 64.90 |
| | Ours | **52.39** | **55.99** | 52.54 | **76.79** | **54.04** | **74.90** | **75.79** | **77.67** | **79.10** | **75.31** | **86.33** | **77.34** | 67.41 | 57.06 | **61.29** | **68.26** |
| | Gain/Loss(%) | +4.80 | +5.81 | -0.61 | +5.40 | +0.18 | +6.98 | +2.89 | +1.30 | +2.80 | +4.53 | +2.21 | +10.99 | -1.59 | -1.51 | +0.15 | +3.36 |
| **CIFAR-100C** | ViT-B/16 | 19.64 | 21.40 | 25.26 | 42.54 | 20.03 | 43.17 | 47.95 | 48.35 | 49.74 | 41.57 | 57.02 | 34.58 | 29.15 | 23.96 | 32.43 | 35.79 |
| | TPT | 17.95 | 19.51 | 27.13 | 43.53 | 20.08 | 42.65 | 48.63 | 49.11 | 49.48 | 42.14 | 57.35 | 33.26 | 31.13 | 27.59 | 32.75 | 36.15 |
| | VTE | 17.96 | 18.72 | 28.17 | 40.38 | 19.60 | 39.50 | 45.33 | 48.24 | 46.87 | 40.73 | 55.31 | 30.04 | 32.47 | 30.35 | 31.45 | 35.01 |
| | Ours | **24.91** | **27.73** | **33.66** | **50.11** | **26.27** | **48.49** | **54.85** | **52.35** | **51.62** | **48.38** | **63.27** | **45.21** | **34.74** | **32.38** | **37.31** | **42.09** |
| | Gain/Loss(%) | +5.27 | +6.33 | +5.49 | +6.58 | +6.19 | +5.32 | +6.22 | +3.24 | +1.88 | +6.24 | +5.92 | +10.63 | +2.27 | +2.03 | +4.56 | +5.94 |
| | ViT-B/32 | 16.23 | 17.83 | 17.57 | 39.07 | 17.63 | 38.55 | 43.81 | 42.32 | 43.46 | 39.71 | 50.32 | 29.34 | 28.74 | 22.85 | 29.42 | 31.79 |
| | TPT | 16.08 | 17.65 | 17.54 | 39.21 | 19.47 | 38.91 | 44.01 | 43.45 | 44.46 | 40.15 | 50.93 | 27.77 | 30.91 | 23.36 | 29.55 | 32.23 |
| | VTE | 16.84 | 18.33 | 18.94 | 39.63 | 22.88 | 39.13 | 43.80 | 44.56 | 44.88 | 39.21 | 49.37 | 28.37 | 34.13 | 26.87 | 30.12 | 33.14 |
| | Ours | **21.35** | **24.71** | **22.32** | **46.26** | **23.07** | **44.64** | **50.12** | **47.23** | **46.88** | **44.92** | **58.55** | **38.52** | **34.56** | **27.73** | **33.19** | **37.60** |
| | Gain/Loss(%) | +4.51 | +6.38 | +3.38 | +6.63 | +0.19 | +5.51 | +6.11 | +2.67 | +2.00 | +4.77 | +7.62 | +9.18 | +0.43 | +0.86 | +3.07 | +4.46 |
| **ImageNet-C** | ViT-B/16 | 11.18 | 12.54 | 12.04 | 23.36 | 15.18 | 24.50 | 22.58 | 32.32 | 29.88 | 35.88 | 54.18 | 17.20 | 12.72 | 30.96 | 33.26 | 24.51 |
| | TPT | 8.48 | 9.46 | 10.20 | 23.98 | 15.16 | 25.10 | 24.00 | 33.94 | 32.12 | 37.08 | 55.64 | 16.54 | 13.68 | 34.06 | 33.58 | 24.87 |
| | VTE | 9.18 | 10.76 | 10.78 | 24.72 | 14.30 | 24.36 | 25.24 | 35.38 | 32.46 | 38.16 | 55.56 | 16.14 | 14.26 | **38.72** | 33.98 | 25.60 |
| | Ours | **19.32** | **21.38** | **19.60** | **26.58** | **21.94** | **30.88** | **29.02** | **36.48** | 32.00 | **40.98** | **56.72** | **26.14** | **23.74** | 37.67 | **38.34** | **30.72** |
| | Gain/Loss(%) | +8.14 | +8.84 | +7.56 | +1.86 | +6.76 | +5.78 | +3.78 | +1.10 | -0.46 | +2.82 | +1.08 | +8.94 | +9.48 | -1.05 | +4.36 | +5.12 |
| | ViT-B/32 | 12.88 | 13.04 | 12.90 | 24.42 | 11.86 | 22.72 | 20.20 | 25.70 | 25.84 | 30.28 | 50.54 | 17.32 | 18.96 | 32.20 | 29.12 | 23.20 |
| | TPT | 12.04 | 12.64 | 12.52 | **25.38** | 12.28 | 22.68 | 20.78 | 26.36 | 26.64 | 30.78 | 51.02 | 16.50 | 19.90 | 33.62 | 30.62 | 23.58 |
| | VTE | 11.96 | 12.32 | 13.44 | 25.06 | 11.70 | 22.58 | 22.40 | 27.38 | 27.02 | 32.28 | **51.52** | 16.84 | 19.94 | 34.80 | 32.82 | 24.14 |
| | Ours | **16.84** | **18.20** | **16.10** | 25.04 | **20.90** | **28.90** | **25.24** | **29.42** | **27.18** | **36.02** | 50.18 | **17.66** | **27.68** | **36.20** | **35.42** | **27.39** |
| | Gain/Loss(%) | +3.96 | +5.16 | +2.66 | -0.34 | +8.62 | +6.18 | +2.84 | +2.04 | +0.16 | +3.74 | -1.34 | +0.34 | +7.74 | +1.40 | +2.60 | +3.25 |

prompt-tuning at test-time, using CLIP. 63 random augmentations are generated for a single test image in the vision space. Based on a threshold, the marginal entropy of confident predictions is minimized to optimize the prompts. In the text space, VTE considers an ensemble of hand-crafted prompts for a test image, to get the final predictions (Eq. 1). Alongside this, they also consider multiple random augmentations of the test image, with frozen CLIP encoders. We also contrast our work against prior TTA methods *i.e.,* TENT (Wang et al., 2021), BN Stats Adapt (BN-1) (Schneider et al., 2020), RPL (Rusak et al., 2021), and SAR (Niu et al., 2022). In essence, we adopt these approaches to use CLIP. Following the guidelines as set by Döbler et al. (2024), for such a setup, we update only the vision encoder $f_{vis}$. The model predictions can then be computed as in Eq. 1. We mention the training details of each TTA method in the Appendix A.2.

**Implementation Details.** We query the text encoder $f_{txt}$ with a general and fixed hand-crafted prompt template "a photo of a <CLS>." for all the datasets, as motivated earlier. We optimize both the vision encoder $f_{vis}$ and text encoder $f_{txt}$. For CIFAR-10C, we use an AdamW optimizer at a learning rate of $10^{-3}$. Similarly, for CIFAR-100C and ImageNet-C, Adam and AdamW optimizers are respectively used, at a fixed learning rate of $5 \times 10^{-4}$, with the model being reset after each task. The batch sizes are set to 200, 200, and 64 for the datasets, following various TTA benchmarks, at a corruption severity level of 5 for each task. As visual backbones, we report results on ViT-B/16 and ViT-B/32 (Dosovitskiy, 2020), where all of our experiments are run on a single NVIDIA RTX A5000 GPU.

**Results on CIFAR-10C, CIFAR-100C, and ImageNet-C.** We present the TTA results in Table 2. Our method significantly improves the mean accuracy across all tasks and datasets for both visual backbones. Unlike TPT (Shu et al., 2022) and VTE (Döbler et al., 2024), which process a single test image at a time and therefore incur substantial time, our approach operates at the batch level. This allows us to leverage the semantic relationships within the batch. By updating the parameters of $f_{vis}$ and $f_{txt}$ through the projection matching loss and enhancing the cosine distance between the class prototypes, our method effectively strengthens the alignment between visual and text features and learns discriminative visual features. This leads to an efficient adaptation to each domain. In Fig. 4, we show certain t-SNE (Van der Maaten & Hinton, 2008) plots of visual and text features and compare them against zero-shot ViT-B/16 for CIFAR-10C. It highlights how our

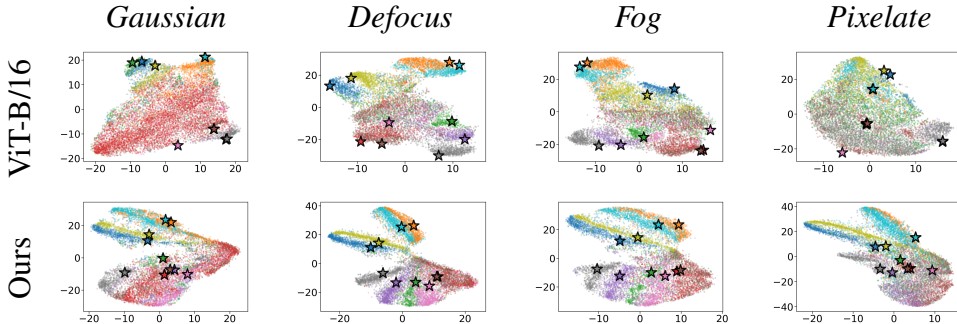

Figure 4: *BAT-CLIP yields more discriminative visual features that exhibit stronger alignment with their corresponding text features.* The t-SNE plots show visual (○) and text (★) features for CIFAR-10C, comparing zero-shot ViT-B/16 with our approach.

approach enhances alignment with text features, fosters better class separation, and forms more distinct clusters, leading to significant improvements compared to zero-shot CLIP. Appendix A.3.5 shows detailed t-SNE plots for CIFAR-10C and CIFAR-100C. Concerning compute time per task, on ImageNet-C, our method takes about 45 s, compared to 40 mins for TPT and 4 mins for VTE. This makes our approach deployment-friendly for quick classification with large improvements.

**Results against prior TTA methods.**

In Table 3, we compare our method with existing TTA approaches, including TENT (Wang et al., 2021), BN-1 (Schneider et al., 2020), RPL (Rusak et al., 2021), and SAR (Niu et al., 2022), all adopted for CLIP. We report mean accuracy across all tasks. While most TTA methods outperform zero-shot CLIP on average, our approach consistently matches or exceeds the performance of these methods across all

Table 3: Mean accuracy (%) for CLIP adopted for TTA approaches at a corruption severity level of 5, using ViT-B/16 and ViT-B/32.

| Dataset | Backbone | ZS | TENT | BN-1 | RPL | SAR | Ours |
|---|---|---|---|---|---|---|---|
| CIFAR-10C | ViT-B/16 | 61.16 | 62.06 | 61.16 | 61.52 | 67.37 | **73.85** |
| | ViT-B/32 | 59.00 | 56.35 | 59.10 | 56.51 | 65.13 | **68.26** |
| CIFAR-100C | ViT-B/16 | 35.79 | 37.96 | 35.79 | 38.47 | 41.19 | **42.09** |
| | ViT-B/32 | 31.79 | 31.61 | 31.78 | 30.00 | **37.81** | 37.60 |
| ImageNet-C | ViT-B/16 | 24.51 | 25.15 | 24.52 | 25.08 | 29.73 | **30.72** |
| | ViT-B/32 | 23.20 | 24.05 | 23.19 | 23.62 | **28.07** | 27.39 |

datasets and visual backbones. Detailed per-task accuracies are provided in Appendix A.3.2. Notably, BN-1 performs similarly to zero-shot CLIP across backbones. This is because BN-1 updates the normalization statistics (mean and variance) based on the test batch, while keeping the model's affine parameters fixed. Since CLIP is pre-trained on diverse distributions, such minor updates to input normalization have little effect on performance. SAR (Niu et al., 2022) performs comparably to our method, except on CIFAR-10C. SAR addresses performance degradation in TTA caused by batch normalization and batch-agnostic layers by filtering noisy test samples, from large gradients, with a stable entropy loss function. While SAR operates solely in the vision space, our approach leverages CLIP's joint vision-language feature space, resulting in superior TTA performance.

## 5.2 ABLATION STUDY AND ANALYSIS

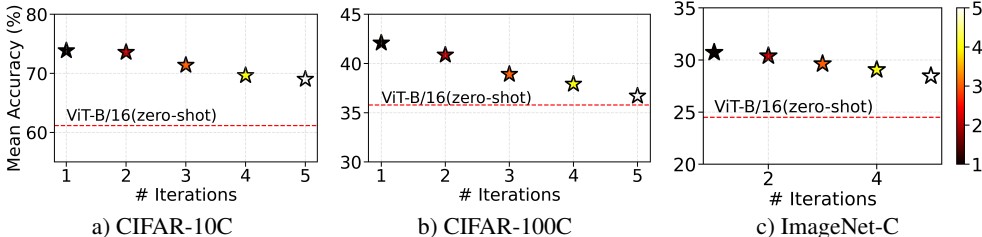

a) CIFAR-10C      b) CIFAR-100C      c) ImageNet-C

Figure 5: **Analysis on increasing # multiple iterations**: Mean accuracy for # iterations, on each test batch, for CIFAR-10C, CIFAR-100C, and ImageNet-C using a ViT-B/16 backbone.

**Adaptation for multiple iterations.** We perform adaptation of models, at test-time, in an online manner *i.e.,* one model update per batch. Here, we analyze an important factor that could affect the

performance of our method - adapting for multiple iterations on a single batch. In Fig. 5, we illustrate the mean accuracy, across all 15 tasks, on the benchmark datasets using the ViT-B/16 visual backbone. Continuous adaptation of the normalization parameters to a single batch can lead to overfitting causing the mean and variance to bias towards the batch and degrading generalization. This could also lead to a decline in the loss of CLIP pre-trained knowledge. In Appendix A.3.4, we report the post-adaptation results back on source test sets - CIFAR10 and CIFAR100. From Fig. 5, our approach still maintains a significant advantage over zero-shot CLIP evaluation. Despite a gradual decline in mean accuracy across datasets, the performance gap remains notably better, particularly on CIFAR-10C and ImageNet-C. This suggests that the proposed loss components contribute to the robustness of our method, preserving its effectiveness even under these challenging conditions.

**Contribution of each loss component.** In Eq. 6, we propose the final loss function to jointly optimize the vision and text encoders of CLIP for TTA. In this study, we validate the effectiveness of each component of the proposed objective. We progressively add each loss component to the final objective and report the mean accuracy across all the tasks in Table 4. The task-wise results are reported in Appendix A.3.3. We observe that the addition of the loss components $\mathcal{L}_{pm}$ and $\mathcal{L}_{sp}$ largely improve model performance than doing simple entropy minimization via $\mathcal{L}_{ent}$. In fact, the addition of $\mathcal{L}_{sp}$ to increase the inter-class separability of prototypes brings larger improvements, proving that $f_{vis}$ produces discriminative features. Interestingly, simple entropy minimization via $\mathcal{L}_{ent}$ achieves better or comparable accuracy than zero-shot CLIP, as the encoders work in synergy through gradient optimization while adapting to the input domain. Hence, the proposed loss components are robust and help in good CLIP adaptation at test-time to image corruptions.

Table 4: Ablation on the impact of different loss components- Mean accuracy (in %) on CIFAR-10C, CIFAR-100C, and ImageNet-C.

| Backbone | CIFAR-10C | CIFAR-100C | ImageNet-C |
|---|---|---|---|
| ViT-B/16 | | | |
| $\mathcal{L}_{ent}$ | 60.65 | 38.17 | 24.03 |
| $\mathcal{L}_{ent}+\mathcal{L}_{pm}$ | 62.60 | 39.32 | 25.21 |
| $\mathcal{L}_{ent}+\mathcal{L}_{pm}+\mathcal{L}_{sp}$ | **73.85** | **42.09** | **30.72** |
| ViT-B/32 | | | |
| $\mathcal{L}_{ent}$ | 54.83 | 33.50 | 21.50 |
| $\mathcal{L}_{ent}+\mathcal{L}_{pm}$ | 59.96 | 35.67 | 21.87 |
| $\mathcal{L}_{ent}+\mathcal{L}_{pm}+\mathcal{L}_{sp}$ | **68.26** | **37.60** | **27.39** |

**Effect of different prompt templates.** In all of our prior experiments, we use a generic prompt template "`a photo of a <CLS>.`" for all of the datasets and methods. Here, we replace this with "relevant" prompt templates to show the independence of such a prompt selection, at test-time, and report the results in Table 5. As seen, the performance gain over zero-shot ViT-B/32 is fairly large for all the prompt templates. Though TPT (Shu et al., 2022) fine-tunes a pre-trained prompt on each test image, and VTE (Döbler et al., 2024) uses an ensemble of prompts, our method is agnostic to the prompt template being used, making it favorable for real-time deployment.

Table 5: Prompt template selection. **+** denotes the accuracy gain over zero-shot ViT-B/32.

| Prompt Template | CIFAR-10C | CIFAR-100C | ImageNet-C |
|---|---|---|---|
| "`a low contrast photo of a <CLS>.`" | 68.53 (**+7.81**) | 37.09 (**+4.97**) | 27.31 (**+3.71**) |
| "`a blurry photo of a <CLS>.`" | 68.84 (**+10.96**) | 36.80 (**+5.33**) | 26.92 (**+3.52**) |
| "`a photo of a big <CLS>.`" | 67.49 (**+10.10**) | 35.79 (**+4.87**) | 25.64 (**+3.29**) |

# 6 CONCLUSION

In this paper, we propose **BAT-CLIP**, a *bimodal* online test-time adaptation framework for CLIP aimed at handling diverse image corruptions simulating real-time environments (Hendrycks & Dietterich, 2019). Our in-depth analysis of CLIP's zero-shot performance under increasing corruption severity reveals significant shortcomings in generalization, highlighting the need for effective adaptation. While prior works on TTA for CLIP have predominantly been *unimodal* focusing on prompt-tuning (Shu et al., 2022) or prompt ensemble (Döbler et al., 2024) with no model updates, our approach encourages learning rich class-separated visual features via vision encoder updates and strengthens alignment between the image class prototype and corresponding text feature via the text encoder updates. Empirical studies, including ablation results, demonstrate the robustness and substantial performance improvements of our approach over existing methods on benchmark corruption datasets - CIFAR-10C, CIFAR-100C, and ImageNet-C, through synergistic CLIP encoder updates.

ETHICS STATEMENT

For our research, we adhere to the highest standards of ethical responsibility. We ensured that the datasets used were publicly available and widely accepted within the research community. Furthermore, no personal data was used or collected during this work, ensuring full compliance with data privacy standards.

REPRODUCIBILITY STATEMENT

To ensure the reproducibility of our work, we relied on open-source pre-trained models, and have provided all the links. We have included the corresponding GitHub repositories with source code, using official or verified reference implementations whenever possible. Detailed hyperparameters are documented and discussed in the Appendix.

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

# A APPENDIX

In this work, we study the problem of test-time adaptation (TTA) of CLIP towards common image corruptions and propose improved schemes for increasing the robustness of CLIP. We put forward a *bimodal* domain adaptation scheme, wherein we exploit the shared feature space of CLIP. In essence, leaning towards a more effective multi-modal learning and adaptation method, we propose loss components that improve alignment between the class-specific visual prototype and corresponding text features via maximizing the projection. We also increase the cosine distance between the class prototypes to enhance discrimination between visual features. In this Appendix, we provide additional insights and experimental results, organized as follows,

1. A.1 offers a detailed discussion of the datasets used, supplemented with visual illustrations.

2. To ensure full transparency, we outline the implementation details of all methods in A.2, including those for prior TTA approaches (Wang et al., 2021; Schneider et al., 2020; Rusak et al., 2021; Niu et al., 2022) adapted for use with CLIP.

3. Section A.3 presents further results and analysis:

   - In A.3.1, we explore the limitations of zero-shot CLIP when using ViT-B/32 and ViT-L/14 backbones under increasing image corruption severity.

   - Table 3 provided mean accuracy for all TTA approaches. A.3.2 expands on this with task-specific accuracies for CIFAR-10C, CIFAR-100C, and ImageNet-C, along with a detailed performance discussion.

   - A.3.3 and A.3.4 present the detailed loss ablation study and post-adaptation zero-shot generalization on source test sets, respectively.

   - Lastly, we include task-wise t-SNE visualizations in A.3.5 for CIFAR-10C and CIFAR-100C, comparing our method against zero-shot CLIP (ViT-B/16), illustrating the effectiveness of **BAT-CLIP**.

## A.1 DATASETS

We employ the **CIFAR-10C**, **CIFAR-100C**, and **ImageNet-C** datasets, for our experiments, as introduced by Hendrycks & Dietterich (2019). Each dataset includes 15 distinct types of image corruptions, referred to as tasks in a test-time adaptation setting, applied to the test sets of CIFAR10, CIFAR100 (Krizhevsky et al., 2009), and ImageNet (Deng et al., 2009). These corruptions are applied at 5 different severity levels, ranging from mild to severe. For each task, CIFAR-10C and CIFAR-100C contain 10,000 test samples, whereas ImageNet-C has 5000 samples.

The image corruptions are categorized into four primary groups: noise, blur, weather, and digital distortions. Noise-based corruptions include *Gaussian*, *Shot*, and *Impulse* noise, which introduce random pixel-level variations. The blur category encompasses *Defocus*, *Glass*, *Motion*, and *Zoom* blur effects, all of which simulate different types of distorted imagery. Weather-related corruptions, such as *Snow*, *Frost*, and *Fog*, replicate environmental conditions that obscure image details. Lastly, digital distortions include effects like *Brightness*, *Contrast*, *Elastic Transform*, *Pixelate*, and *JPEG* compression, which reflect various forms of post-processing or compression artifacts that degrade image quality.

These corruption types, as proposed by Hendrycks & Dietterich (2019), provide a comprehensive framework for assessing model robustness, which has been and is still being studied (Hendrycks & Gimpel, 2016; Metzen et al., 2017; Papernot et al., 2016; Subbaswamy et al., 2021; Liu et al., 2024). Their ability to emulate real-world image degradation scenarios is advantageous, allowing for a more realistic evaluation of a model's robustness. We provide corruption visualizations, via an image example, in Fig. 6. For further inspection, we urge the readers to check out Hendrycks & Dietterich (2019).

## A.2 IMPLEMENTATION DETAILS

In this subsection, we summarise the implementation details of all the baseline methods that have been mentioned in the main paper, including ours. We build our approach on the open-source code

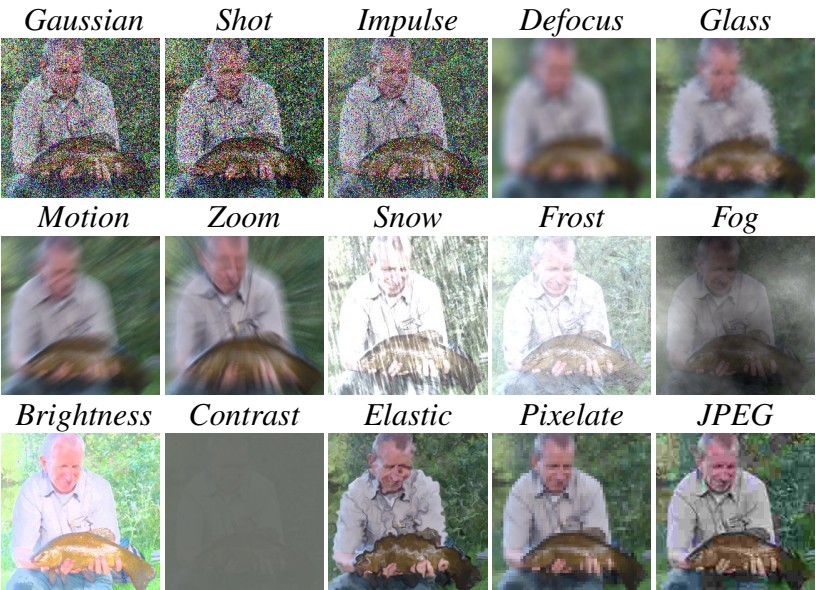

Figure 6: We provide visualizations of an image from ImageNet-C (Hendrycks & Dietterich, 2019) for different corruption types, at an image severity level of 5.

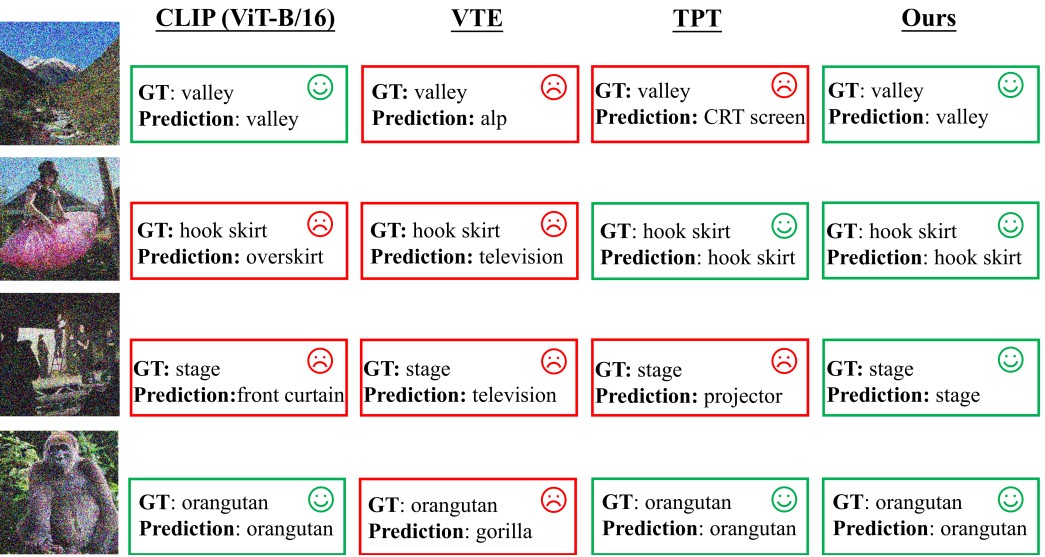

Figure 7: Comparison of classification predictions across various methods (Zero-shot CLIP (ViT-B/16), VTE (Döbler et al., 2024), TPT (Shu et al., 2022), and Ours) on ImageNet-C samples with *Gaussian* noise. Each row illustrates a unique example, displaying the ground truth (GT) label alongside the predictions from each method. Correct predictions are highlighted in green, while incorrect ones are marked in red. Our approach demonstrates enhanced robustness and higher accuracy, especially in challenging image corruption conditions.

base [2]. a standard TTA benchmark codebase, that also houses the hyperparameters and training details of all the prior TTA methods. CLIP-like models are used as provided by OpenCLIP. Only the vision encoder is updated for online TTA methods (Wang et al., 2021; Schneider et al., 2020; Rusak et al., 2021; Niu et al., 2022) adopted for CLIP.

---

[2]https://github.com/mariodoebler/test-time-adaptation/tree/main

**BAT-CLIP (Ours)**: For domain-specific test adaptation, we conducted experiments using ViT-B/16 and ViT-B/32 (Dosovitskiy, 2020) as the vision backbones. For CIFAR-10C, both the vision encoder ($f_{vis}$) and text encoder ($f_{txt}$) were updated using the AdamW optimizer with a learning rate of $10^{-3}$. Similarly, for CIFAR-100C and ImageNet-C, we employed the Adam optimizer and AdamW optimizer, respectively, with a learning rate of $5 \times 10^{-4}$. The batch size $\mathcal{B}$ used was set to 200 for CIFAR-10C and CIFAR-100C, and 64 for ImageNet-C. Throughout, the prompt template is fixed to "a photo of a <CLS>.".

**TPT (Shu et al., 2022)**: For each test image, 63 augmentations are generated based on random resized crops, yielding a batch of 64 images, in addition to the original test image. The prompt/context vectors are initialized based on "a photo of a <CLS>." and tokenized using pre-trained CLIP weights. The confidence threshold is set to 10% *i.e.,* the marginal entropy over the 10% confident samples is minimized. For all the datasets, we follow their core implementation and optimize the prompt vectors using an AdamW optimizer with a learning rate of $5 \times 10^{-3}$.

**VTE (Döbler et al., 2024)**: In VTE, an ensemble of different prompt templates is considered based on the idea of Radford et al. (2021). An example of templates includes "a photo of a <CLS>.", "a sketch of a <CLS>.", "a painting of a <CLS>.", etc. The prompt templates are then averaged. On the vision side, similar to TPT (Shu et al., 2022), a batch of random augmentation is created for a test image.

**TENT (Wang et al., 2021)**: We follow all the hyperparameters that are provided by TENT in their official implementation [3]. To update the vision encoder, we use Adam as the optimizer with a learning rate of $10^{-3}$ for CIFAR-10C and CIFAR-100C. For ImageNet-C, we update using SGD with a learning rate of $25 \times 10^{-5}$.

**BN Stats Adapt (BN-1) (Schneider et al., 2020)**: BN-1 recomputes the statistics of the batch normalization layers based on the input test batch, consisting of corruption of a certain domain. Hence, this requires no model updates.

**RPL (Rusak et al., 2021)**: We use an Adam optimizer with a learning rate of 1e-3 for CIFAR-10C and CIFAR-100C. For ImageNet-C, the update rule is SGD with a learning rate of $5 \times 10^{-4}$. To compute the generalized cross-entropy loss, $q$ is set to 0.8 for all the datasets.

**SAR (Niu et al., 2022)**: The training details/hyperparameters for SAR are the same as RPL (Rusak et al., 2021) for CIFAR-10 and CIFAR-100. For ImageNet-C, the learning rate is set to $25 \times 10^{-5}$ with an SGD update rule. The entropy threshold $E_0$ is 0.4xln(C), where C is the number of classes. $\rho$ is set to a default of 0.05. The moving average factor is 0.9 for $e_m$ and $e_0$ is set to 0.2. We completely follow the implementation details as outlined in their main paper.

### A.3 ADDITIONAL RESULTS

#### A.3.1 ZERO-SHOT PERFORMANCE ANALYSIS OF VIT-B/32 AND VIT-L/14

In Section 3 of the main paper, we analyze and evaluate the zero-shot performance of ResNet-101 (RN101) (He et al., 2016) and ViT-B-16 (Dosovitskiy, 2020) and conclude that such CLIP backbones are extremely sensitive, in terms of classification accuracy, to increasing severity levels of image corruption. This could be a major concern in situations involving real-time deployment of CLIP. Here, we present a similar analysis in Fig. 8, using ViT-B/32 and ViT-L/14 as backbones. Our analysis, from the main paper, carries forward. To summarise, irrespective of the CLIP visual backbone, the robustness towards image corruption is limited. The classification performance degrades with an increase in the severity of corruption in an image.

#### A.3.2 RESULTS AGAINST PRIOR TTA METHODS

In Table 3, we present the mean accuracy, across all the tasks, for prior TTA methods (Wang et al., 2021; Schneider et al., 2020; Rusak et al., 2021; Niu et al., 2022) adapted for CLIP vs ours. Here, we provide the fine-grained results *i.e.,* the task-wise mean accuracy to demonstrate the efficacy of our method (the higher the better). The results are reported in Tables 6, 7, and 8. Across all

---

[3]https://github.com/DequanWang/tent

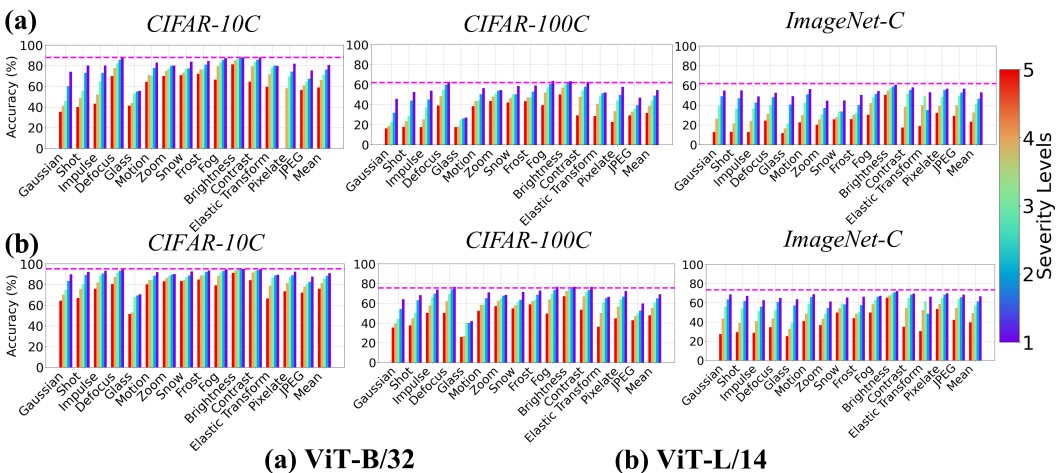

**(a) ViT-B/32**  **(b) ViT-L/14**

Figure 8: Task-wise mean accuracy (%) of zero-shot CLIP across different corruption severity levels. [Top]: ViT-B/32 backbone. [Bottom]: ViT-L/14 backbone. The **dashed lines** indicate the performance of zero-shot CLIP (w/ respective visual backbones) on the corresponding source datasets.

Table 6: Mean accuracy (%) on CIFAR-10C - TTA mean accuracy of the 15 corruptions (tasks) at a severity level of 5, using ViT-B/16 and ViT-B/32. We contrast our results against zero-shot backbones, TENT (Wang et al., 2021), BN-1 (Schneider et al., 2020), RPL (Rusak et al., 2021), and SAR (Niu et al., 2022).

| | Method | Gaussian | Shot | Impulse | Defocus | Glass | Motion | Zoom | Snow | Frost | Fog | Brightness | Contrast | Elastic | Pixelate | JPEG | Mean |
|---|---|---|---|---|---|---|---|---|---|---|---|---|---|---|---|---|---|
| CIFAR-10C | ViT-B/16 | 37.92 | 41.70 | 54.42 | 71.75 | 40.89 | 67.93 | 73.62 | 73.89 | 77.35 | 70.22 | 84.45 | 62.36 | 53.81 | 47.65 | 59.43 | 61.16 |
| | TENT | 15.49 | 18.28 | 38.12 | 81.59 | 21.73 | 76.32 | 82.35 | **84.62** | 82.19 | 80.60 | **91.83** | 80.55 | 63.52 | 58.57 | 54.71 | 62.03 |
| | BN-1 | 37.99 | 41.73 | 54.40 | 71.70 | 40.91 | 67.92 | 73.61 | 73.89 | 77.41 | 70.26 | 84.47 | 62.29 | 53.82 | 47.60 | 59.40 | 61.16 |
| | RPL | 15.47 | 17.43 | 40.73 | **81.76** | 20.08 | 69.89 | **82.93** | 84.43 | 83.19 | **81.84** | 91.80 | 79.42 | 64.89 | 54.07 | 54.90 | 61.52 |
| | SAR | 47.98 | 53.60 | 60.56 | 74.30 | 47.56 | 73.15 | 76.43 | 77.91 | 79.88 | 75.66 | 86.79 | 71.62 | 58.34 | 62.03 | 64.71 | 67.37 |
| | Ours | **61.13** | **64.09** | **65.76** | 80.51 | **54.96** | 80.65 | 81.94 | 83.04 | 84.19 | 80.84 | 88.95 | **82.15** | 69.16 | 62.68 | 67.64 | **73.85** |
| | ViT-B/32 | 35.47 | 39.94 | 43.23 | 69.95 | 41.43 | 64.50 | 70.13 | 70.85 | 72.33 | 66.66 | 81.37 | 64.57 | 59.69 | 48.28 | 56.62 | 59.00 |
| | TENT | 20.09 | 23.45 | 34.47 | 69.85 | 23.01 | 39.79 | 60.35 | 76.83 | 77.49 | 76.07 | **88.88** | **81.38** | 65.35 | 57.01 | 51.19 | 56.35 |
| | BN-1 | 35.58 | 40.07 | 43.16 | 69.98 | 41.50 | 64.51 | 70.19 | 70.80 | 72.34 | 66.66 | 81.38 | 64.51 | 59.66 | 48.16 | 56.58 | 59.10 |
| | RPL | 15.89 | 19.08 | 34.04 | **77.84** | 18.72 | 41.22 | 62.39 | **78.17** | 78.86 | **76.31** | 88.83 | 81.15 | **68.98** | 54.19 | 51.91 | 56.51 |
| | SAR | 50.28 | 54.12 | 49.65 | 73.08 | 51.98 | 71.17 | 74.65 | 73.73 | 75.22 | 70.99 | 84.25 | 72.08 | 63.93 | 51.57 | 60.32 | 65.13 |
| | Ours | **52.39** | **55.99** | **52.54** | 76.79 | **54.04** | 74.90 | 75.79 | 77.67 | **79.10** | 75.31 | 86.33 | 77.34 | 67.41 | **57.06** | 61.29 | 68.26 |

Table 7: Mean accuracy (%) on CIFAR-100C - TTA mean accuracy of the 15 corruptions (tasks) at a severity level of 5, using ViT-B/16 and ViT-B/32. We contrast our results against zero-shot backbones, TENT (Wang et al., 2021), BN-1 (Schneider et al., 2020), RPL (Rusak et al., 2021), and SAR (Niu et al., 2022).

| | Method | Gaussian | Shot | Impulse | Defocus | Glass | Motion | Zoom | Snow | Frost | Fog | Brightness | Contrast | Elastic | Pixelate | JPEG | Mean |
|---|---|---|---|---|---|---|---|---|---|---|---|---|---|---|---|---|---|
| CIFAR-100C | ViT-B/16 | 19.64 | 21.40 | 25.26 | 42.54 | 20.03 | 43.17 | 47.95 | 48.35 | 49.74 | 41.57 | 57.02 | 34.58 | 29.15 | 23.96 | 32.43 | 35.79 |
| | TENT | 7.60 | 8.21 | 8.33 | 51.81 | 7.95 | **52.45** | 55.34 | 54.16 | 36.17 | 50.92 | 65.63 | **54.51** | 36.52 | **43.99** | 35.81 | 37.96 |
| | BN-1 | 19.57 | 21.39 | 25.26 | 42.46 | 20.08 | 43.19 | 47.98 | 48.44 | 49.70 | 41.69 | 57.00 | 34.47 | 29.21 | 23.93 | 32.47 | 35.79 |
| | RPL | 6.44 | 7.09 | 7.09 | 52.16 | 11.81 | 52.33 | **55.50** | 54.20 | 38.83 | **51.99** | 66.07 | 54.45 | 36.86 | 42.83 | **39.45** | 38.47 |
| | SAR | **25.30** | 27.19 | 32.78 | 47.12 | 23.42 | 47.16 | 51.70 | 51.94 | **52.48** | 48.77 | 61.54 | 44.50 | 32.26 | 33.67 | 38.06 | 41.19 |
| | Ours | 24.91 | **27.73** | **33.66** | 50.11 | **26.27** | 48.49 | 54.85 | 52.35 | 51.62 | 48.38 | 63.27 | 45.21 | 34.74 | 32.23 | 37.31 | **42.09** |
| | ViT-B/32 | 16.23 | 17.83 | 17.57 | 39.07 | 17.63 | 38.55 | 43.81 | 42.32 | 43.46 | 39.71 | 50.32 | 29.34 | 28.74 | 22.85 | 29.42 | 31.79 |
| | TENT | 5.53 | 7.64 | 6.85 | **49.60** | 4.47 | 48.45 | 52.35 | 49.77 | 26.77 | 37.50 | 63.05 | **50.53** | 13.89 | 27.00 | 30.80 | 31.61 |
| | BN-1 | 16.20 | 17.82 | 17.55 | 39.06 | 17.68 | 38.59 | 43.83 | 42.30 | 43.37 | 39.59 | 50.38 | 29.36 | 28.78 | 22.86 | 29.40 | 31.78 |
| | RPL | 4.50 | 5.80 | 9.61 | 50.26 | 4.43 | **48.88** | **52.61** | 50.27 | 22.36 | 25.34 | **63.36** | 50.31 | 9.10 | 18.65 | **34.53** | 30.00 |
| | SAR | 24.63 | 27.14 | 21.25 | 44.57 | 22.98 | 43.95 | 48.40 | 48.01 | **47.76** | 44.85 | 57.76 | 42.11 | 32.69 | **28.02** | 33.08 | **37.81** |
| | Ours | 21.35 | 24.71 | **22.32** | 46.26 | **23.07** | 44.64 | 50.12 | 47.23 | 46.88 | **44.92** | 58.55 | 38.52 | **34.56** | 27.73 | 33.19 | 37.60 |

the backbones and datasets, we see that our method **BAT-CLIP** achieves the best or comparable performance against all the baseline TTA approaches adopted for CLIP.

Table 8: Mean accuracy (%) on ImageNet-C - TTA mean accuracy of the 15 corruptions (tasks) at a severity level of 5, using ViT-B/16 and ViT-B/32. We contrast our results against zero-shot backbones, TENT (Wang et al., 2021), BN-1 (Schneider et al., 2020), RPL (Rusak et al., 2021), and SAR (Niu et al., 2022).

| | Method | Gaussian | Shot | Impulse | Defocus | Glass | Motion | Zoom | Snow | Frost | Fog | Brightness | Contrast | Elastic | Pixelate | JPEG | Mean |
|---|---|---|---|---|---|---|---|---|---|---|---|---|---|---|---|---|---|
| ImageNet-C | ViT-B/16 | 11.18 | 12.54 | 12.04 | 23.36 | 15.18 | 24.50 | 22.58 | 32.32 | 29.88 | 35.88 | 54.08 | 17.20 | 12.72 | 30.96 | 33.26 | 24.51 |
| | TENT | 5.14 | 5.70 | 7.44 | 25.22 | 19.34 | 26.80 | 24.16 | 33.56 | 30.42 | 37.74 | 54.24 | 22.50 | 13.90 | 35.02 | 36.08 | 25.15 |
| | BN-1 | 11.12 | 12.52 | 11.98 | 23.32 | 15.22 | 24.52 | 22.68 | 32.30 | 30.00 | 35.82 | 54.04 | 17.26 | 12.72 | 31.06 | 33.24 | 24.52 |
| | RPL | 9.04 | 10.04 | 10.96 | 24.40 | 17.40 | 26.28 | 23.76 | 32.70 | 30.62 | 36.64 | 54.04 | 19.38 | 13.24 | 33.14 | 34.60 | 25.08 |
| | SAR | 17.96 | 20.46 | **20.68** | 25.72 | **23.04** | 29.52 | 26.04 | 34.92 | **32.74** | 39.00 | 55.00 | **27.14** | 19.64 | 36.66 | 37.50 | 29.73 |
| | Ours | **19.32** | **21.38** | 19.60 | **26.58** | 21.94 | **30.88** | **29.02** | **36.48** | 32.00 | **40.98** | **56.72** | 26.14 | **23.74** | **37.68** | **38.34** | **30.72** |
| | ViT-B/32 | 12.88 | 13.04 | 12.90 | 24.42 | 11.86 | 22.72 | 20.20 | 25.70 | 25.84 | 30.28 | 50.54 | 17.32 | 18.96 | 32.20 | 29.12 | 23.20 |
| | TENT | 9.18 | 8.50 | 10.42 | 26.02 | 15.72 | 26.06 | 21.64 | 27.12 | 26.18 | 31.60 | 50.58 | 22.28 | 20.12 | 34.06 | 31.30 | 24.05 |
| | BN-1 | 12.84 | 13.02 | 12.82 | 24.44 | 11.84 | 22.72 | 20.24 | 25.70 | 25.86 | 30.18 | 50.50 | 17.38 | 18.92 | 32.30 | 29.10 | 23.19 |
| | RPL | 11.68 | 10.98 | 12.10 | 25.62 | 13.24 | 23.98 | 20.84 | 26.32 | 26.12 | 30.86 | 50.62 | 19.30 | 19.48 | 33.14 | 29.92 | 23.62 |
| | SAR | **19.82** | **20.36** | **20.92** | 25.78 | 20.40 | 28.34 | 23.10 | 28.12 | **28.38** | **34.74** | **51.10** | 24.60 | 24.38 | **36.54** | 34.40 | **28.07** |
| | Ours | 16.84 | 18.20 | 16.10 | 25.04 | **20.90** | **28.90** | **25.24** | **29.42** | 27.18 | 36.02 | 50.18 | 17.66 | **27.58** | 36.20 | **35.42** | 27.39 |

Table 9: Task-wise loss ablation results (accuracy) on CIFAR-10C, CIFAR-100C, and ImageNet-C.

| | Method | Gaussian | Shot | Impulse | Defocus | Glass | Motion | Zoom | Snow | Frost | Fog | Brightness | Contrast | Elastic | Pixelate | JPEG | Mean |
|---|---|---|---|---|---|---|---|---|---|---|---|---|---|---|---|---|---|
| CIFAR-10C | ViT-B/16 | | | | | | | | | | | | | | | | |
| | $\mathcal{L}_{ent}$ | 14.62 | 17.29 | 49.25 | **81.06** | 20.23 | 74.27 | 81.10 | **84.25** | 81.93 | 80.93 | **91.86** | 78.92 | 53.09 | 51.18 | 49.79 | 60.65 |
| | $\mathcal{L}_{ent}+\mathcal{L}_{pm}$ | 16.58 | 19.89 | 42.69 | 79.45 | 23.41 | 77.03 | 80.95 | 81.74 | 78.45 | 80.66 | 90.52 | **82.55** | 62.56 | **64.35** | 58.16 | 62.60 |
| | $\mathcal{L}_{ent}+\mathcal{L}_{pm}+\mathcal{L}_{sp}$ | **61.13** | **64.09** | **65.76** | 80.51 | **54.96** | 80.65 | **81.94** | 83.04 | **84.19** | 80.84 | 88.95 | 82.15 | **69.16** | 62.68 | **67.64** | **73.85** |
| | ViT-B/32 | | | | | | | | | | | | | | | | |
| | $\mathcal{L}_{ent}$ | 16.30 | 19.83 | 33.86 | 67.22 | 18.55 | 42.53 | 63.87 | 75.21 | 75.16 | 74.75 | **89.40** | 81.28 | 67.41 | 46.06 | 50.45 | 54.83 |
| | $\mathcal{L}_{ent}+\mathcal{L}_{pm}$ | 26.31 | 29.38 | 37.02 | 75.16 | 40.52 | 56.37 | 72.26 | 76.18 | 77.14 | 74.32 | 87.42 | 77.07 | 66.69 | 49.57 | 54.03 | 59.96 |
| | $\mathcal{L}_{ent}+\mathcal{L}_{pm}+\mathcal{L}_{sp}$ | **52.39** | **55.99** | **52.54** | **76.79** | **54.04** | **74.90** | **75.79** | **77.67** | **79.10** | **75.31** | 86.33 | 77.34 | 67.41 | **57.06** | **61.29** | **68.26** |
| CIFAR-100C | ViT-B/16 | | | | | | | | | | | | | | | | |
| | $\mathcal{L}_{ent}$ | 7.71 | 10.05 | 11.52 | 49.42 | 12.49 | **49.36** | 53.79 | **54.11** | 50.76 | **49.92** | 64.32 | **47.07** | 33.40 | **38.63** | 39.95 | 38.17 |
| | $\mathcal{L}_{ent}+\mathcal{L}_{pm}$ | 12.26 | 12.62 | 13.14 | 48.90 | 26.22 | 48.99 | 53.10 | 53.10 | **52.43** | 49.44 | 63.36 | 46.78 | 33.27 | 37.77 | 38.36 | 39.32 |
| | $\mathcal{L}_{ent}+\mathcal{L}_{pm}+\mathcal{L}_{sp}$ | **24.91** | **27.73** | **33.66** | **50.11** | **26.27** | 48.49 | **54.85** | 52.35 | 51.62 | 48.38 | 63.27 | 45.21 | **34.74** | 32.38 | 37.31 | **42.09** |
| | ViT-B/32 | | | | | | | | | | | | | | | | |
| | $\mathcal{L}_{ent}$ | 9.91 | 10.71 | 9.90 | **47.61** | 7.65 | **45.96** | 51.30 | 49.25 | 38.23 | 44.90 | 60.04 | 41.53 | 29.47 | 21.67 | **35.22** | 33.50 |
| | $\mathcal{L}_{ent}+\mathcal{L}_{pm}$ | 12.37 | 15.19 | 10.52 | 46.78 | 13.33 | 45.42 | 50.16 | 48.76 | **48.73** | 46.69 | 59.33 | 42.64 | 33.09 | **28.82** | 33.28 | 35.67 |
| | $\mathcal{L}_{ent}+\mathcal{L}_{pm}+\mathcal{L}_{sp}$ | **21.35** | **24.71** | **22.32** | 46.26 | **23.07** | 44.64 | 50.12 | 47.23 | 46.88 | 44.92 | 58.55 | 38.52 | **34.56** | 27.73 | 33.19 | **37.60** |
| ImageNet-C | ViT-B/16 | | | | | | | | | | | | | | | | |
| | $\mathcal{L}_{ent}$ | 0.90 | 1.06 | 1.16 | **29.12** | 13.02 | **32.14** | 27.34 | 35.32 | 11.14 | 40.92 | **56.90** | 23.78 | 7.78 | 39.62 | 40.22 | 24.03 |
| | $\mathcal{L}_{ent}+\mathcal{L}_{pm}$ | 0.90 | 1.16 | 1.30 | 28.90 | 17.04 | 31.56 | 26.24 | 36.26 | 12.22 | **42.12** | 57.92 | **30.34** | 10.36 | **40.66** | **41.20** | 25.21 |
| | $\mathcal{L}_{ent}+\mathcal{L}_{pm}+\mathcal{L}_{sp}$ | **19.32** | **21.38** | **19.60** | 26.58 | **21.94** | 30.88 | **29.02** | **36.48** | **32.00** | 40.98 | 56.72 | 26.14 | **23.74** | 37.68 | 38.34 | **30.72** |
| | ViT-B/32 | | | | | | | | | | | | | | | | |
| | $\mathcal{L}_{ent}$ | 2.54 | 2.06 | 2.40 | **28.06** | 8.16 | **30.38** | 21.68 | 23.14 | 10.68 | 21.22 | **52.68** | 25.12 | 16.46 | **40.06** | **37.90** | 21.50 |
| | $\mathcal{L}_{ent}+\mathcal{L}_{pm}$ | 2.82 | 2.28 | 2.70 | 27.62 | 10.10 | 28.74 | 18.36 | 24.16 | 11.38 | 27.78 | 52.40 | **25.66** | 17.62 | 39.16 | 37.22 | 21.87 |
| | $\mathcal{L}_{ent}+\mathcal{L}_{pm}+\mathcal{L}_{sp}$ | **16.84** | **18.20** | **16.10** | 25.04 | **20.90** | 28.90 | **25.24** | **29.42** | **27.18** | **36.02** | 50.18 | 17.66 | **27.58** | 36.20 | 35.42 | **27.39** |

### A.3.3 DETAILED RESULTS FROM THE LOSS ABLATION STUDY

In Table 4 of the main paper, we provide ablation of loss components *i.e.,* the mean accuracy across all the tasks for ViT-B/16 and ViT-B/32 on the benchmarks corruption datasets. Here, we provide additional task-wise accuracy in Table 9. Indeed, the addition of loss components $\mathcal{L}_{pm}$ and $\mathcal{L}_{sp}$ to entropy loss $\mathcal{L}_{ent}$ indeed helps in improving the robustness of CLIP to different corruption tasks.

Table 10: Zero-shot performance on CIFAR10 (source) after adaptation of BAT-CLIP on a task.

| | Method | Gaussian | Shot | Impulse | Defocus | Glass | Motion | Zoom | Snow | Frost | Fog | Brightness | Contrast | Elastic | Pixelate | JPEG | Zero-Shot |
|---|---|---|---|---|---|---|---|---|---|---|---|---|---|---|---|---|---|
| | ViT-B/16 | | | | | | | | | | | | | | | | 90.1 |
| | Ours | 84.51 | 84.29 | 88.69 | 88.48 | 86.44 | 86.46 | 87.04 | 91.38 | 91.01 | 90.22 | 90.87 | 88.12 | 88.13 | 74.74 | 87.48 | 87.19 (mean) |
| | ViT-B/32 | | | | | | | | | | | | | | | | 88.3 |
| | Ours | 67.41 | 68.28 | 84.23 | 80.50 | 75.37 | 79.75 | 78.55 | 87.67 | 86.36 | 85.83 | 90.04 | 80.89 | 81.56 | 82.74 | 82.36 | 80.77 (mean) |

Table 11: Zero-shot performance on CIFAR100 (source) after adaptation of BAT-CLIP on a task.

| Method | Gaussian | Shot | Impulse | Defocus | Glass | Motion | Zoom | Snow | Frost | Fog | Brightness | Contrast | Elastic | Pixelate | JPEG | Zero-Shot |
|---|---|---|---|---|---|---|---|---|---|---|---|---|---|---|---|---|
| ViT-B/16 Ours | 67.09 | 67.12 | 67.08 | 70.31 | 63.70 | 66.83 | 70.12 | 70.10 | 68.19 | 69.55 | 71.05 | 66.49 | 65.13 | 59.98 | 67.60 | 66.6 67.36 (mean) |
| ViT-B/32 Ours | 45.99 | 46.90 | 60.86 | 63.62 | 57.73 | 59.59 | 61.66 | 66.21 | 61.50 | 63.61 | 66.39 | 57.05 | 62.13 | 62.82 | 65.37 | 62.3 60.09 (mean) |

### A.3.4 POST-ADAPTATION RESULTS ON SOURCE TEST SETS

Thanks to the natural language supervision and also due to the pre-training on large amounts of (image, text) pairs, CLIP has shown strong generalization capabilities. However, for an efficient adaptation to a downstream task, fine-tuning the full model is infeasible due to large model updates. The primary reason is the loss of useful pre-trained knowledge of CLIP, which could eventually lead to overfitting to a downstream task. In our *bimodal* test-adaptation scheme, **BAT-CLIP**, we take inspiration from Sreenivas & Biswas (2024) and update only the *LayerNorm parameters* of the CLIP encoders to a specific corruption task, which makes it parametric-efficient. Now, with continual adaptation to a corruption task, it gets difficult to preserve CLIP's pre-trained knowledge. Then, a natural question arises -

*Given that CLIP has been adapted to a specific corruption task, will the zero-shot generalization still hold back on its source test set?*

In this crucial experiment, we challenge our **BAT-CLIP** and evaluate its zero-shot generalization performance back on the source test set, to check the preservation of pre-trained CLIP Knowledge. After the adaptation of CLIP on each corruption task, we report the adapted model's zero-shot performance on its corresponding source test set. We report results for CIFAR-10C and CIFAR-100C in Tables 10 and 11, using ViT-B/16 and ViT-B/32 backbones. For all of the results, we use the prompt template "a photo of a <CLS>.". As an example, for CIFAR-10C, upon adaptation of CLIP to *Gaussian noise* following our approach, we report the adapted model's zero-shot accuracy on its source test set - CIFAR10 test set.

From Table 10, we observe that, on average, there is a 2.91% drop in accuracy compared to a zero-shot evaluation using pre-trained CLIP ViT-B/16. Similarly, for ViT-B/32, we see a drop of about 7.53% in mean accuracy. In Table 11, for CIFAR-100C using a ViT-B/16 backbone, we see an improvement of 0.76% in mean accuracy.

On the whole, we conclude that since the adaptation for a task happens over multiple test batches, the zero-shot performance back on the source data largely depends on the distribution of the image corruption. Overall, ViT-B/16 visual backbones preserve larger amounts of CLIP pre-trained knowledge. This proves the effectiveness of our method **BAT-CLIP**, on average.

### A.3.5 T-SNE VISUALIZATIONS ON CIFAR-10C AND CIFAR-100C

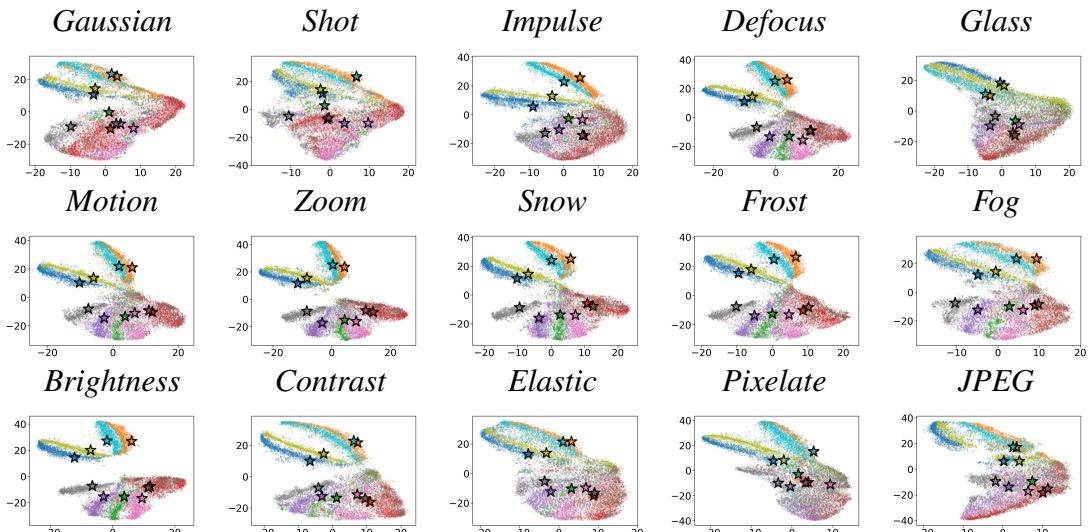

Figure 9: BAT-CLIP (w/ ViT-B/16): The t-SNE plots show visual (○) and text (★) features for CIFAR-10C.

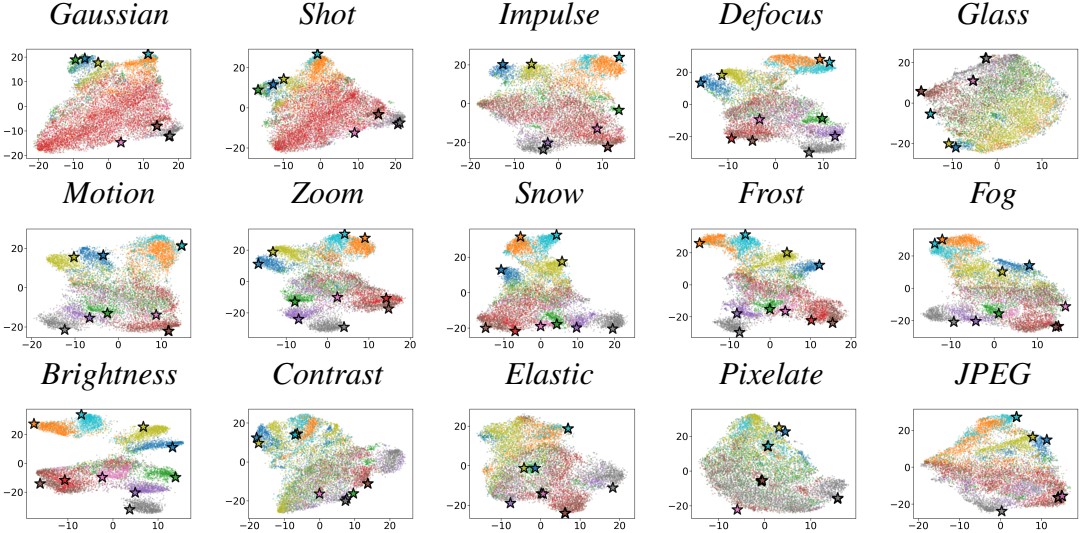

Figure 10: Zero-shot ViT-B/16: The t-SNE plots show visual (○) and text (★) features for CIFAR-10C.

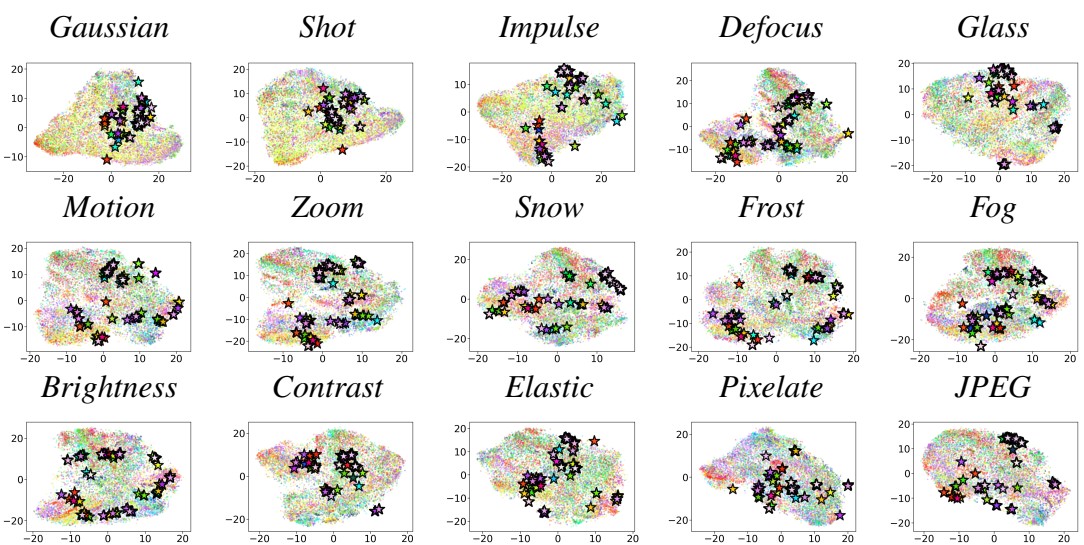

Figure 11: BAT-CLIP (w/ ViT-B/16): The t-SNE plots show visual (○) and text (★) features for CIFAR-100C.

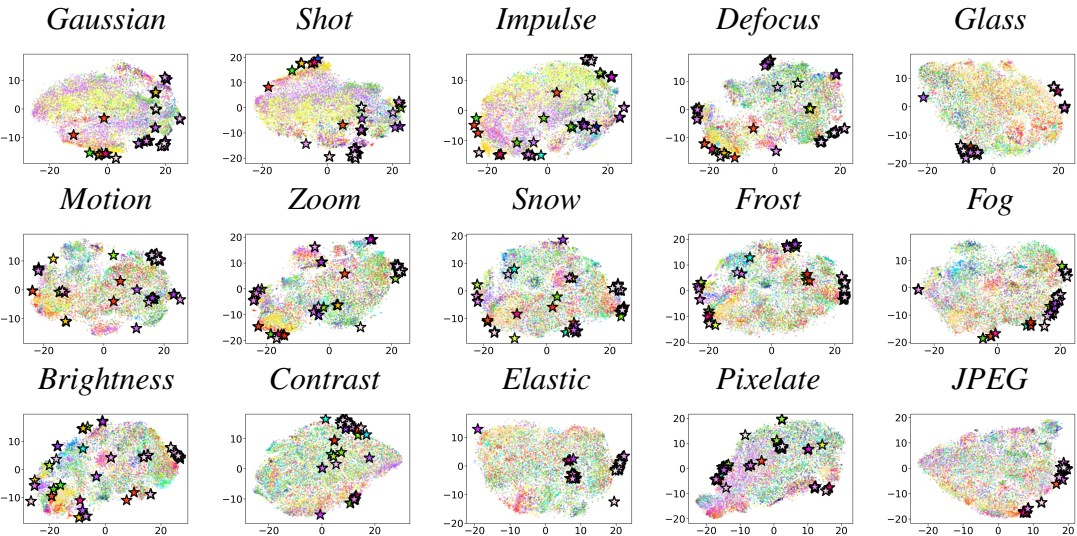

Figure 12: Zero-shot ViT-B/16: The t-SNE plots show visual (○) and text (★) features for CIFAR-100C.