# OpenReview forum: "BAT-CLIP: Bimodal Test-Time Adaptation for CLIP"
_ICLR.cc/2025/Conference — ICLR 2025 Conference Withdrawn Submission_

### Official Review · Reviewer_MucK · 2024-10-30

**Soundness:** 3
**Presentation:** 3
**Contribution:** 3
**Rating:** 6
**Confidence:** 5

**Summary:**

The authors address the problem of TTA in the context of VLMs. Specifically, they propose a bimodal TTA to improve the performance of CLIP for domain shifted datasets. They improve the performance by encouraging the text and image prototypes to match. And they also enhance the discrimination between the class prototypes. The layernorm parameters in both vision and text encoder are updated during test time.

**Strengths:**

- The proposed method is simple and intuitive to understand. The paper is well presented and is easy to understand.
- The motivation and experimental analysis on the performance degradation of CLIP under corruptions is well presented.
- The method is very efficient compared to prior methods like TPT.
- The experimental results show significant improvements compared to previous methods which are even more computationally expensive.

**Weaknesses:**

- **Ablation:** Please report Zero shot CLIP results followed by the loss components. It appears that $L_{tent}$ sometimes worse than Zero shot CLIP for some cases. What happens if you only use $L_{tent}$, $L_{pm}$ and $L_{sp}$ individually. A better ablation study is where you study all the loss combinations, which I encourage the authors to present.
- **Bimodal Adapatation:** As the primary motivation and difference from prior works is the need to do bimodal adaptation, experiments demonstrating the effectiveness of bimodal adaptation is missing. Some experiments like: What if you use $L_{pm}$ without updating the text encoder? How do you show bimodal adaptation is better than unimodal update of LN parameters of Vision encoder?, could be done to highlight the role of bimodal adaptation.
- **Lacking strong experimental protocol:** All the experiments are done on only corruption benchmarks. How would BAT-CLIP perform in other domain shifts in datasets like DomainNet, VisDA comprising of Cartoon, Sketch kind of domains and ImageNet-variants like IN-R/Sketch/V2/A? While not a necessity, a stronger set of baseline methods would be to compare with more recent works like[4,5,6].
- **Projection matching loss:**: In eqn 2: What happens in low batchsize setting or when some classes are absent in the batch? How would the prototypes be computed for this loss? The prototypes are computed for every batch? Also, performing experiments on varying batchsizes would further demonstrate you method's effectiveness. Why is this a projection based loss where unnormalized protypes are used and not cosine similarity based, as in $L_{sp}$? Is there any specific reason for this choice?

**Questions:**

- In Section 4, Optimization, you mention "For every new task, we reset the model parameters of CLIP following TENT (Wang et al., 2021) since our goal is to adapt to a single domain in an online manner." What do you mean by a task here? Is each corruption treated independently? Like single domain TTA protocol?
- **CTTA Scenario:** How would BAT-CLIP perform in long range test sequences and continual TTA scenarios as studied in (CoTTA[1], RMT[2]).
- **TSNE plots:** It is well known that there still exists a huge modality gap in CLIP feature space; Image-image features, text-text features are closer compared to image-text features. So, the text features form a cluster, away from image features, irrespective of the classes. This is studied extensively in several works[3,4]. So how are these plots obtained where the text features seem to be close to image features.
- **BN-1 Results:** All the experiments are done on ViT architectures which do not have Batch-Normalization layers. This makes no sense. Do you mean Layer Normalization. If so, LN layers behave the same way training and testing time. So LN-1 would mean Zero shot CLIP evaluation only.
- **TENT and SAR Baselines:** How are these adapted to CLIP? The objectives can be used. But as there are no BN layers in ViT, what parameters are updated? The comment "SAR addresses performance degradation in TTA caused by batch normalization and batch-agnostic layers by filtering noisy test samples, from large gradients, with a stable entropy loss function" makes no sense in ViT based TTA. Please clarify these.
- **BN statistics based observations:** Again, in the section 'Adaptation for multiple iterations', the authors mention "Continuous adaptation of the normalization parameters to a single batch can lead to over-fitting causing the mean and variance to bias towards the batch and degrading generalization". This in the content of ViT architecture without BN layers needs to be justified.

---

### Official Review · Reviewer_NHUN · 2024-11-02

**Soundness:** 2
**Presentation:** 3
**Contribution:** 2
**Rating:** 5
**Confidence:** 4

**Summary:**

This paper presents BAT-CLIP, a bimodal test-time adaptation method designed to enhance the robustness of the CLIP model against image corruption. BAT-CLIP adapts both CLIP’s visual and text encoders by updating LayerNorm parameters. During adaptation, in addition to minimizing entropy loss, two additional loss functions leverage pseudo-labels: $L_{pm}$ maximizes the projection of class prototypes with their corresponding text features, while $L_{sp}$ increases the cosine distance between the class prototypes. The method is evaluated on corrupted image datasets including CIFAR-10C, CIFAR-100C, and ImageNet-C.

**Strengths:**

- The paper is well-written and easy to understand.
- Empirical results demonstrate the usefulness of the two additional loss functions.

**Weaknesses:**

- The claim that the proposed method is the first to perform a bimodal test-time adaptation of CLIP for classification tasks is imprecise, see, e.g., Section 4 of [1]. Moreover, since maximizing the similarity between visual and text features (e.g., CLIP original training objective) and increasing inter-class separability are common practices, I do not consider the approach as particularly novel.
- Experimental evaluation should be improved. (1) Some highly related SOTA methods [2, 3, 4, 5]  lack detailed discussion and comparison.  (2) Additional datasets are needed to further validate the effectiveness of the approach, such as ImageNet-V2, ImageNet-A, ImageNet-R, and ImageNet-Sketch, which are commonly used in the TTA of CLIP [3].
- Contributions are not fully supported by experimental evidence and should be clarified. (1) In the ablation study (Table 4), it is unclear whether the significant improvement with the two additional loss terms can also be achievable without updating the text encoder. (2) To support the claim of efficient adaptation, comparisons with previous baselines, especially [3],  in terms of FLOPs or at least forward/backward computation times are needed. It is important given that updating the text encoder seems computationally intensive, as all text prompts seem to be processed again through the text encoder at each update step.

[1] Döbler, Mario, et al. "A Lost Opportunity for Vision-Language Models: A Comparative Study of Online Test-Time Adaptation for Vision-Language Models." CVPR Workshop.

[2] Sreenivas, Manogna, and Soma Biswas. "Effectiveness of Vision Language Models for Open-World Single Image Test Time Adaptation." *arXiv preprint arXiv:2406.00481* (2024).

[3] Niu, Shuaicheng, et al. "Test-Time Model Adaptation with Only Forward Passes." ICML, 2024.

[4] Ma, Xiaosong, et al. "SwapPrompt: Test-Time Prompt Adaptation for Vision-Language Models." NeurIPS, 2023.

[5] Zhang, Jingyi, et al. "Historical Test-Time Prompt Tuning for Vision Foundation Models." NeurIPS, 2024.

**Questions:**

- Please find the weaknesses.

---

### Official Review · Reviewer_7diB · 2024-11-03

**Soundness:** 3
**Presentation:** 3
**Contribution:** 3
**Rating:** 8
**Confidence:** 3

**Summary:**

The paper proposes BAT-CLIP, a bimodal test-time adaptation method designed to improve the robustness of the CLIP model against common image corruptions during testing. The key idea is to jointly adapt both the visual and text encoders of CLIP by exploiting its shared feature space. The proposed adaptation is realized through LayerNorm parameter updates and leverages two novel loss components: a projection matching loss (to enhance image-text alignment) and a separation loss (to improve class separability).

**Strengths:**

1. This paper proposes a bimodal test-time adaptation method that effectively utilizes both visual and text modalities, enhancing the adaptation process and improving alignment between image and text features.
2. The authors conduct extensive experiments on benchmark datasets, demonstrating that BAT-CLIP achieves state-of-the-art results in TTA for CLIP, with notable accuracy improvements across multiple datasets.
3. The paper is well organized and easy to follow.

**Weaknesses:**

1. Limited benchmarking against prior methods: The current paper primarily compares its approach to TTA methods from 2020 to 2022. It would provide a clearer understanding of BAT-CLIP's performance within the broader TTA landscape if the authors were to include comparisons with a wider range of state-of-the-art TTA methods in 2023 and 2024.
2. Limited Exploration of Text Encoder's Role: While the text encoder is adapted alongside the vision encoder, the paper doesn't deeply explore cases where the text encoder might contribute disproportionately to misalignment. For instance, how does BAT-CLIP handle noisy or ambiguous text class descriptions during adaptation? This could be important in real-world applications where text inputs may not always be clean or well-formed.
3. Lack of hyperparameter sensitivity analysis: The method introduces projection matching loss and separation loss, but no details are given on how sensitive the method is to hyperparameters such as the weighting between these two losses, or the cosine distance threshold for class prototypes. This sensitivity needs to be explored experimentally to ensure the method is robust.
4. Limited justification for LayerNorm adaptation: The paper chooses to adapt the LayerNorm parameters of CLIP’s encoders, as inspired by the other method. However, there is no strong theoretical or empirical justification for why adapting LayerNorm parameters is the optimal choice for both the vision and text encoders. It’s not clear if other layers might benefit from adaptation as well.

**Questions:**

1. Further justification for updating only the LayerNorm parameters: The paper should provide a more thorough explanation or any experimental validation of why only the LayerNorm parameters of the CLIP encoder are updated for the specific corrupted task, such as a comparison between using only LayerNorm updates and fully fine-tuning the model versus selectively fine-tuning other layers.
2. Enhance ablation and comparative experiments: It is suggested that the authors refine their ablation and comparative experiments in accordance with the feedback provided above to strengthen the overall analysis.
3. Impact of corruption types: It would be useful to see how BAT-CLIP performs on specific types of corruption (e.g., noise vs. blur vs. weather effects). Do certain corruptions benefit more from the bimodal adaptation, and if so, why? A more granular analysis would provide deeper insights into the strengths and weaknesses of BAT-CLIP.

---

### Official Review · Reviewer_PDsN · 2024-11-05

**Soundness:** 2
**Presentation:** 2
**Contribution:** 2
**Rating:** 3
**Confidence:** 4

**Summary:**

This work propose to improve CLIP’s robustness to common image corruptions through the proposed bimodal test-time adaptation method. The proposed method adapts the visual encoders and strengthen the alignment between image and text features using three losses, computed using pseudo-labels, and the corresponding text feature. The adaptation is performed only on the layer normalization parameters.

**Strengths:**

1. Updating both the image encoder and text encoder for CLIP is good.
2. The experimental results seem promising.
3. However, the experimental setting is unfair and this work seems violate the test-time adaptation setting.

**Weaknesses:**

The method design and experiment have fatal problems.
1. The test-time adaptation task assumes that there is no any class label for test images, because we need to use the model to predict the label for test images. However, this work directly leverage the groundtruth class label of test images to train the model, e.g., ``we compute the mean feature of all the support visual features constituting a class c. . yˆ refers to the predicted labels computed via Eq. 1 and v¯c is the class prototype of class c.'' in L333-334. It is obvious that the method is trained using the class label of test images.
2. The test-time adaptation task typically assume that the model can process limited samples during test-time adaptation, e.g., one test image, or up to 16 images. However, this work takes 200 images as a mini-batch for TTA, as shown in L419, ``The batch sizes are set to 200, 200, and 64 for the datasets''. In contrast, other methods, e.g., TPT (Shu et al., 2022) and VTE (Dobler et al., 2024) process a single test image at a time.
3. The novelty and contribution is limited. The proposed losses are the popular entropy minimization, projection loss and contrastive loss. The model adaptation method is the classical normalization layer adaptation.
4. The authors claim that ''image class prototype are computed using pseudo-labels'' in the abstract. However, I cannot find any ``pseudo-labels'' in the method part. It is obvious that the authors did not produce and use the pseudo labels. Instead, the authors directly use the groundtruth class labels of test images to train the model.

**Questions:**

Why does authors use the groundtruth class labels alongside the test images to train the model for predicting class labels of the test images?

---

### Note · Authors · 2024-11-12

I have read and agree with the venue's withdrawal policy on behalf of myself and my co-authors.